# A Variational Perspective on Diffusion-Based Generative Models and Score Matching

**Chin-Wei Huang,  Jae Hyun Lim,  Aaron Courville**
University of Montreal & Mila
{chin-wei.huang, jae.hyun.lim, aaron.courville}@umontreal.ca

## Abstract

Discrete-time diffusion-based generative models and score matching methods have shown promising results in modeling high-dimensional image data. Recently, Song et al. (2021) show that diffusion processes that transform data into noise can be reversed via learning the score function, *i.e.* the gradient of the log-density of the perturbed data. They propose to plug the learned score function into an inverse formula to define a generative diffusion process. Despite the empirical success, a theoretical underpinning of this procedure is still lacking. In this work, we approach the (continuous-time) generative diffusion directly and derive a variational framework for likelihood estimation, which includes continuous-time normalizing flows as a special case, and can be seen as an infinitely deep variational autoencoder. Under this framework, we show that minimizing the score-matching loss is equivalent to maximizing a lower bound of the likelihood of the plug-in reverse SDE proposed by Song et al. (2021), bridging the theoretical gap.

## 1 Introduction

Generative modeling can be thought of as inverting an inference process. If the inference process is invertible, then one can focus on transforming the data into a tractable distribution (Dinh et al., 2016). If the inference process is deterministic yet non-invertible, one could learn to invert it stochastically (Dinh et al., 2019; Nielsen et al., 2020). Most generally, both inference and generation can be stochastic. This is known as the variational autoencoder (Kingma & Welling, 2014; Rezende et al., 2014, VAE).

Under the variational framework, one has a lot of flexibility in choosing the generative and inference models. Recent work on diffusion-based modeling (Sohl-Dickstein et al., 2015; Ho et al., 2020) can be thought of as removing one degree of freedom, by freezing the inference path. The inference model is a fixed discrete-time Markov chain, that slowly transforms the data into a tractable prior, such as the standard normal distribution. The generative model is another Markov chain that is trained to revert this process iteratively. Diffusion-based models have been shown to perform remarkably well on image synthesis (Dhariwal & Nichol, 2021), rivaling the performance of state-of-the-art Generative Adversarial Networks (Brock et al., 2018).

Song et al. (2021) connect diffusion-based model and *score matching* (Hyvärinen & Dayan, 2005), by looking at the stochastic differential equation (SDE) associated with the inference process. They realize that the dynamic of the inference process can be inverted if one has access to the score function of the perturbed data, by solving another SDE reversed in time. They then propose to learn the score function of the inference process and substitute the approximate score into the formula of the reverse SDE to obtain a generative model. We call the resulting generative model the plug-in reverse SDE.

35th Conference on Neural Information Processing Systems (NeurIPS 2021).

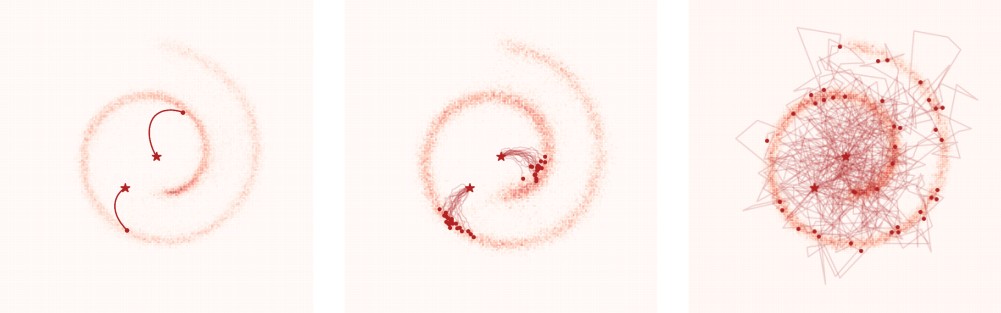

Figure 1: Three special cases of generative SDEs. The stars indicate the initial values, followed by some random sample paths. *Left*: trained with no diffusion $\sigma = 0$ (*i.e.* neural ODE). *Middle*: trained with some fixed diffusion $\sigma > 0$. *Right*: trained with a fixed inference process, $f$ and $g$ (*i.e.* the plug-in reverse SDE).

Conceptually simple as this learning procedure may seem, little is known about how the score matching loss relates to the plug-in reverse SDE. In this paper, we propose a variational framework suitable for likelihood estimation for general generative diffusion processes, and use this framework to connect score matching with maximum likelihood. We do so by combining two important theorems in stochastic calculus: the *Feynman-Kac formula* for representing the marginal density of the generative diffusion as an expectation (Section 3), and the *Girsanov theorem* for performing inference in function space (Section 4). We derive a functional evidence lower bound that consistently extends discrete-time diffusion models to have infinite depth, *i.e.* the number of layers goes to infinity (Section 5). Finally, by reparameterizing our generative and inference SDEs, we obtain a training objective equivalent to minimizing the (implicit) score matching loss (Section 6). Our theory suggests that by matching the score, one actually maximizes a lower bound on the log marginal density of the plug-in reverse SDE, laying a theoretical foundation for this learning procedure. We further generalize our result to a family of *marginal-equivalent* plug-in reverse SDEs, including an equivalent ODE as a limiting case.

**Notation:** We use $(Y_s, s)$ to denote the inference process (where $Y_0$ is the data), and $(X_t, t)$ to denote the generative process (where $X_0$ is a random variable following an unstructured prior). We use $s$ and $t$ to distinguish the two directions, and always integrate the differential equations from $0$ to $T > 0$ (different from the literature, where sometimes one might see integration from $T$ to $0$). $\hat{B}_s$ and $B_t$ denote the Brownian motions associated with the inference and generative SDEs, respectively. $B'_s$ is a reparameterization of $\hat{B}_s$ (see Section 4). $q(y, s)$ and $p(x, t)$ denote the probability density functions of $Y_s$ and $X_t$, respectively. We let $\boldsymbol{s}_\theta$ denote a time-indexed parameterized function that will be used to approximate the score $\nabla \log q(y, s)$. $\nabla$ is the gradient wrt the spatial variable ($x$ or $y$, which we sometimes call position), $\partial_t$, $\partial_s$ and $\partial_{x_i}$ are partial derivatives, and $H_*$ denotes Hessian.

## 2 Background

Assume $Y_0$ follows the data distribution $q(y, 0)$, and $Y_s$ satisfies the Itô SDE (Øksendal, 2003)

$$\mathrm{d}Y = f(Y, s)\,\mathrm{d}s + g(Y, s)\,\mathrm{d}\hat{B}_s, \tag{1}$$

where $f$ and $g$ are chosen such that the density $q(y, s)$ will converge to some tractable prior $p_0$ as $s \to T$. Following Song et al. (2021), we assume $g$ is position-independent. It is possible to find a "reverse" SDE, whose marginal density evolves according to $q(y, s)$, reversed in time, for example[1]

$$\mathrm{d}X = (gg^\top \nabla \log q(X, T - t) - f)\,\mathrm{d}t + g\,\mathrm{d}B_t. \tag{2}$$

If $X_0 \sim p_0$, then the density $p(x, t)$ of $X_t$ is equal to $q(x, T - t)$. This means that if we have access to the score function $\nabla \log q$, we can solve the above SDE to obtain $X_T \overset{d}{=} Y_0$. Song et al. (2021)

---

[1]See Appendix G for a family of equivalent (reverse) SDEs indexed by some parameter $\lambda$, of which equation (2) is a special case with $\lambda = 0$.

| Method | Loss |
|--------|------|
| $\mathcal{L}_{\text{ESM}}$ | $\frac{1}{2}\mathbb{E}[||\boldsymbol{s}_\theta(Y_s,s) - \nabla\log q(Y_s)||_\Lambda^2]$ |
| $\mathcal{L}_{\text{ISM}}$ | $\mathbb{E}[\frac{1}{2}||\boldsymbol{s}_\theta(Y_s,s)||_\Lambda^2 + \nabla\cdot(\Lambda^\top\boldsymbol{s}_\theta)]$ |
| $\mathcal{L}_{\text{SSM}}$ | $\mathbb{E}[\frac{1}{2}||\boldsymbol{s}_\theta(Y_s,s)||_\Lambda^2 + v^\top\nabla(\Lambda^\top\boldsymbol{s}_\theta)v]$ |
| $\mathcal{L}_{\text{DSM}}$ | $\frac{1}{2}\mathbb{E}[||\boldsymbol{s}_\theta(Y_s,s) - \nabla\log q(Y_s|Y_0)||_\Lambda^2]$ |

Table 1: Score matching losses. $v$ follows the Rademacher distribution.

| F-K | F-P |
|-----|-----|
| $v(y,\varsigma)$ | $p(y,T-\varsigma)$ |
| $c(y,\varsigma)$ | $-\nabla\cdot\mu(y,T-\varsigma)$ |
| $b(y,\varsigma)$ | $-\mu(y,T-\varsigma)$ |
| $\eta(y,\varsigma)$ | $\sigma(T-\varsigma)$ |
| $g(y)$ | $p_0(y)$ |

Table 2: Feynman-Kac coefficients.

propose to approximate the score via a parameterized score function $\boldsymbol{s}_\theta$ by minimizing

$$\int_0^T \mathbb{E}_{Y_s}\left[\frac{1}{2}||\boldsymbol{s}_\theta(Y_s,s) - \nabla\log q(Y_s,s)||_{\Lambda(s)}^2\right]\mathrm{d}s$$

where the expectation in the integral is known as the explicit score matching (ESM) loss $\mathcal{L}_{\text{ESM}}$, and $\Lambda(s)$ is a positive definite matrix[2] that serves as a weighting function for the overall loss. $\mathcal{L}_{\text{ESM}}$ is not immediately useful, since we do not have access to the ground truth score $\nabla\log q$. A few alternative losses can be used, which are all equal to one another up to a constant, including implicit score matching (Hyvärinen & Dayan, 2005, ISM), sliced score matching (Song et al., 2020, SSM), and denoising score matching (Vincent, 2011, DSM). The losses are summarized in Table 1, and are related through the following identity (see Appendix A for the derivation):

$$\mathcal{L}_{\text{ESM}} - \frac{1}{2}\mathcal{I}(q(y_s,s)) = \mathcal{L}_{\text{ISM}} = \mathcal{L}_{\text{SSM}} = \mathcal{L}_{\text{DSM}} - \frac{1}{2}\mathbb{E}_{Y_0}[\mathcal{I}(q(y_s|y_0))], \tag{3}$$

where $\mathcal{I}(q) = \mathbb{E}[||\nabla\log q||_\Lambda^2]$ is a constant. After training, Song et al. (2021) plug $\boldsymbol{s}_\theta$ into (2) to define a generative model. We refer to this SDE as the *plug-in reverse SDE*. The plug-in reverse SDE has been demonstrated to have impressive empirical results, but a theoretical underpinning of this learning framework is still lacking. For example, it is unclear how the training objective (minimizing the score matching loss) relates to the sampling procedure, *e.g.* whether the probability distribution induced by the plug-in reverse SDE gets closer to the data distribution in the sense of any statistical divergence or metric. We seek to answer the following question in this paper: *How will minimizing the score-matching loss impact the plug-in reverse SDE?* We first provide a framework to estimate the likelihood of generative SDEs, and then get back to this question in Section 6.

## 3 Marginal density and stochastic instantaneous change of variable

Let $X_t$ be a diffusion process solving the following Itô SDE[3]:

$$\mathrm{d}X = \mu(X,t)\,\mathrm{d}t + \sigma(X,t)\,\mathrm{d}B_t \tag{4}$$

with the initial condition $X_0 \sim p_0$, which induces a family of densities $X_t \sim p(\cdot,t)$. We use this SDE as the generative SDE, and we are interested in $\log p(x,T)$ for maximum likelihood. The density $p(x,t)$ follows the *Kolmogorov forward* (or the *Fokker Planck*) *equation*:

$$\partial_t p(x,t) = -\sum_j \partial_{x_j}[\mu_j(x,t)\,p(x,t)] + \sum_{i,j}\partial^2_{x_i,x_j}[D_{ij}(x,t)\,p(x,t)] \tag{5}$$

with the initial value $p(\cdot,0) = p_0(\cdot)$, where $D = \frac{1}{2}\sigma\sigma^T$ is the diffusion matrix. We can expand the Fokker Planck and rearrange the terms to obtain

$$\partial_t p(x,t) = \left[-\nabla\cdot\mu(x,t) + \sum_{i,j}\partial^2_{x_i,x_j}D_{ij}(x,t)\right]p(x,t) +$$

$$\sum_i\left[-\mu_i(x,t) + 2\sum_j\partial_{x_j}D_{ij}(x,t)\right]\partial_{x_i}p(x,t) + \sum_{i,j}D_{ij}(x,t)\partial^2_{x_i,x_j}p(x,t) \tag{6}$$

[2]We use this matrix to induce a Mahalanobis norm $||x||_\Lambda^2 := x^\top\Lambda x$, which will be used in Section 6.

[3]For generality, we use the notation $\mu$ and $\sigma$ to describe a generative SDE, which will be set to $g^2\boldsymbol{s}_\theta - f$ and $g$ when we come back to the discussion of the plug-in reverse SDE in Section 6.

so that all coefficients of the same order are grouped together. For simplicity, we assume the diffusion term $\sigma$ is independent of $x$ throughout the paper. Then (6) reduces to

$$\partial_t p(x,t) = -\left(\nabla \cdot \mu(x,t)\right) p(x,t) - \mu(x,t)^\top \nabla p(x,t) + D(t) : H_p(x,t) \tag{7}$$

where : denotes the Frobenius inner product between matrices. Even with this simplification, solving (7) is not trivial. Fortunately, we can estimate this quantity using the *Feynman-Kac formula*, which tells us that the solution of certain second-order linear partial differential equations have a probabilistic representation.

**Theorem 1** (**Feynman-Kac representation**, Chapter 5.7 of Karatzas & Shreve (2014)). *Let $T > 0$. Let $y$ and $\varsigma$ be the spatial and temporal arguments to the function $v \in C^{2,1}(\mathbb{R}^d \times [0,T])$ solving*

$$\partial_\varsigma v + cv + b^\top \nabla v + A : H_v = 0 \tag{8}$$

*with the terminal condition $v(y,T) = h(y)$, where $A = \frac{1}{2}\eta\eta^\top$ for some matrix-valued function $\eta(y,\varsigma)$. Under the assumption stated in Appendix B, if $B'_s$ is a Brownian motion and $Y_s$ solves*

$$\mathrm{d}Y = b(Y,s)\,\mathrm{d}s + \eta(Y,s)\,\mathrm{d}B'_s, \tag{9}$$

*with the initial datum $Y_\varsigma = y$, then*

$$v(y,\varsigma) = \mathbb{E}\left[h(Y_T)\exp\left(\int_\varsigma^T c(Y_s,s)\,\mathrm{d}s\right)\,\middle|\,Y_\varsigma = y\right]. \tag{10}$$

To estimate the density $p(\cdot,T)$ of (7), we can apply the change of variable $p(x,t) := v(x,T-t)$ by letting the Feynman-Kac (F-K) coefficients correspond to their Fokker-Planck (F-P) counterparts according to Table 2. This way, solving (8) backward is equivalent to solving (7) forward, and we have the following representation of the marginal density at $T$:

$$p(x,T) = \mathbb{E}\left[p_0(Y_T)\exp\left(\int_0^T -\nabla \cdot \mu(Y_s,T-s)\,\mathrm{d}s\right)\,\middle|\,Y_0 = x\right], \tag{11}$$

where $Y_s$ is a diffusion process solving

$$\mathrm{d}Y = -\mu(Y,T-s)\,\mathrm{d}s + \sigma(T-s)\,\mathrm{d}B'_s. \tag{12}$$

**Remark 1** (**Marginalization**). This representation can be interpreted as a mixture of continuous time flows. Assume a sample path of the Brownian motion is given, and we are interested in how the density evolves following the dynamic (4). In the infinitesimal setting, it can been seen as applying the invertible map $x \mapsto x + \mu(x,t)\Delta t + \sigma(t)\Delta B_i$, where $\Delta B_i := B_{(i+1)\Delta t} - B_{i\Delta t}$ is the Brownian increment. Since the diffusion term is independent of the spatial variable, it can be seen as a constant additive transformation, which is volume preserving, so it will not be taken into account when computing the change of density. The only contribution to the change of density will be from the log-determinant of the Jacobian of $\mathrm{id} + \mu\Delta t$, which means we can simply apply the instantaneous change of variable formula (Chen et al., 2018). This will be the conditional density given the entire $\{B_t : t \geq 0\}$, and marginalizing it out results in the expectation in (11). See Appendix C for details.

Our framework also works with the general case where $\sigma$ depends on $x$, but the formulae need to be adapted to account for the spatial partial derivatives. See Appendix D for the derivation.

## 4   Inferring latent Brownian motion

As our goal is to estimate likelihood, we would like to compute the log density value using (11). However, this involves integrating out all possible Brownian paths, which is intractable. To resolve this, we view the Brownian motion as a latent variable, and perform inference by assigning higher probability to sample paths that are more likely to generate the observation. One can view this as a VAE, except we have an infinite dimensional latent variable.

Formally, let $(\Omega, \mathcal{F}, \mathbb{P})$ be the underlying probability space for which $B'_s$ is a Brownian motion. Suppose $\mathbb{Q}$ is another probability measure on $(\Omega, \mathcal{F})$ *equivalent* to $\mathbb{P}$; that is, $\mathbb{P}$ and $\mathbb{Q}$ are similar in

the sense that they have the same measure zero sets. This allows us to apply the change-of-measure trick and lower bound the log-likelihood with a finite quantity using Jensen's inequality:

$$\log p(x, T) \geq \mathbb{E}_{\mathbb{Q}} \left[ \log \frac{\mathrm{d}\mathbb{P}}{\mathrm{d}\mathbb{Q}} + \log p_0(Y_T) - \int_0^T \nabla \cdot \mu \, \mathrm{d}s \, \middle| \, Y_0 = x \right]. \tag{13}$$

Note that $\frac{\mathrm{d}\mathbb{P}}{\mathrm{d}\mathbb{Q}}$ is the *Radon-Nikodym* derivative of $\mathbb{P}$ wrt $\mathbb{Q}$. When both measures are absolutely continuous wrt a third measure, say Lebesgue, then the derivative can be expressed as the ratio of the two densities. However, since we are dealing with an infinite dimensional space, we are immediately faced with the following problems:

1. Is there a measure $\mathbb{Q}$ (equiv. to $\mathbb{P}$) for which $\frac{\mathrm{d}\mathbb{P}}{\mathrm{d}\mathbb{Q}}$ can be easily computed, or at least numerically approximated?

2. Can we find a reparameterization (similar to the Gaussian reparameterization) of $B'_s$ under the new law $\mathbb{Q}$ to estimate the gradient needed for training?

We resort to the *Girsanov theorem*, which describes a general framework for dealing with the change of measure of Gaussian random variables under additive perturbation. It allows us to consider the law of a diffusion process as $\mathbb{Q}$. See Appendix E for an explanation using the more familiar notion of probability densities.

**Theorem 2** (**Girsanov theorem**, Theorem 8.6.3 of Øksendal (2003))**.** *Let $\hat{B}_s$ be an Itô process solving*

$$\mathrm{d}\hat{B}_s = -a(\omega, s) \, \mathrm{d}s + \mathrm{d}B'_s, \tag{14}$$

*for $\omega \in \Omega$, $0 \leq s \leq T$ and $\hat{B}_0 = 0$, where $a(\omega, s)$ satisfies the Novikov's condition $\mathbb{E}\left[\exp\left(\frac{1}{2}\int_0^T a^2 \, \mathrm{d}s\right)\right] < \infty$. Then $\hat{B}_s$ is a Brownian motion wrt $\mathbb{Q}$ where*

$$\frac{\mathrm{d}\mathbb{Q}}{\mathrm{d}\mathbb{P}}(\omega) := \exp\left(\int_0^T a(\omega, s) \cdot \mathrm{d}B'_s - \frac{1}{2}\int_0^T \|a(\omega, s)\|_2^2 \, \mathrm{d}s\right). \tag{15}$$

Equation (14) provides a standarization formula of $B'_s$ under $\mathbb{Q}$, which means we can "invert" it to reparameterize $B'_s$. This leads to the following lower bound.

**Theorem 3** (**Continuous-time ELBO**)**.** *Let $\mathbb{Q}$ be defined via the density (15). Then the RHS of (13) can be rewritten as*

$$\mathbb{E}\left[-\frac{1}{2}\int_0^T \|a(\omega, s)\|_2^2 \, \mathrm{d}s + \log p_0(Y_T) - \int_0^T \nabla \cdot \mu \, \mathrm{d}s \, \middle| \, Y_0 = x\right] =: \mathcal{E}^\infty, \tag{16}$$

*where the expectation is taken wrt the Brownian motion $\hat{B}_s$, and $Y_s$ solves*[4]

$$\mathrm{d}Y = (-\mu + \sigma a) \, \mathrm{d}s + \sigma \mathrm{d}\hat{B}_s. \tag{17}$$

We call $Y_s$ solving (17) the inference SDE, and $\mathcal{E}^\infty$ the continuous-time ELBO (CT-ELBO).

**Remark 2** (**Computation**)**.** This lower bound can be numerically estimated by using any black box SDE solver, by augmenting the dynamic of $y$ with the accumulation of $\|a\|^2$ and $\nabla \cdot \mu$. Computing the divergence term $\nabla \cdot \mu$ directly can be expensive, but it can be efficiently estimated using the Hutchinson trace estimator (Hutchinson, 1989) along with reverse-mode automatic differentiation, similar to Grathwohl et al. (2018). As the parameters of both the generative and inference models are decoupled from the random variable $\hat{B}_s$, their gradients can be estimated via the reparameterization trick (Kingma & Welling, 2014; Rezende et al., 2014). Furthermore, backpropagation can be computed using an adjoint method with a constant memory cost (Li et al., 2020).

**Remark 3** (**Drift** $a$)**.** (i) In general, the drift term of the approximate posterior to the latent Brownian motion can be amortized, so that it will encode the information of individual datum $x$. (ii) The regularization $\|a\|^2$ ensures that $a$ is kept close to $0$, since it represents the deviation of the measure

---

[4]Note that $\mu$ and $\sigma$ run backward in time from $T$, whereas $a$ runs forward.

it induces (*i.e.* $\mathbb{Q}$) from the classical Wiener measure (which is a centered Gaussian measure). (iii) When the diffusion coefficient $\sigma$ is 0, the inference SDE reduces to the reverse dynamic of the generative ODE, and if $a \equiv 0$ in this case, the lower bound is tight. (iv) There is generally no constraint on the form of $a(\omega, s)$, so one can potentially augment it with additional dimensions to have a non-Markovian inference SDE. For simplicity, we let the inference SDE be a Markovian model, i.e. $a = a(y, s)$. This is justified by the following theorem.

**Theorem 4** (**Variational gap and optimal inference SDE**). *The variational gap can be written as*

$$\log p(x, T) - \mathcal{E}^\infty = \int_0^T \mathbb{E}\left[||a(\omega, s) - \sigma^\top \nabla \log p(Y_s, T - s)||^2\right] \, \mathrm{d}s. \tag{18}$$

*In particular, $\mathcal{E}^\infty = \log p(x, T)$ if and only if $a(\omega, s)$ can be written as $a(\omega, s) = a(Y_s(\omega), s)$ for almost every $s \in [0, T]$ and $\omega \in \Omega$, and $a(y, s) = \sigma^\top \nabla \log p(y, T - s)$ almost everywhere.*

**Remark 4** (**Variational gap**). Even though the inference SDE seemingly takes a simple form, it is sufficiently flexible in that this type of variational problem can be generally solved by taking the supremem over all progressively measurable processes $a(\omega, s)$ (Boué et al., 1998). In fact, the above theorem shows that $\mathcal{E}^\infty = \log p(x, T)$ if and only if $a(y, s) = \sigma^\top \nabla \log p(y, T - s)$. This means a non-amortized Markovian inference process is powerful enough.

# 5 Infinitely deep hierarchical VAE

Before we make the connection to score matching, we formally address the common belief that "diffusion models can be viewed as the continuous limit of hierarchical VAEs" (Tzen & Raginsky, 2019), and show that the CT-ELBO consistently extends their discrete-time counterpart. We do so by inspecting the ELBO of a hierarchical VAE defined as discretized[5] generative and inference SDEs. We assume the generative model (*i.e.* the decoder) follows the transitional distributions

$$p(x_{i+1}|x_i) = \mathcal{N}(x_{i+1}; \tilde{\mu}_i(x_i), \tilde{\sigma}_i^2) \tag{19}$$

$$\tilde{\sigma}_i^2 = \Delta t \sigma^2(i\Delta t), \tag{20}$$

where $\Delta t = T/L$ is the step size and $L$ is the number of layers. For the inference model (*i.e.* the encoder), we assume

$$q(x_i|x_{i+1}) = \mathcal{N}(x_i; \hat{\mu}_{i+1}(x_{i+1}), \hat{\sigma}_{i+1}^2) \tag{21}$$

$$\hat{\mu}_i(x) = x + \Delta t(-\mu(x, i\Delta t) + \sigma(i\Delta t)a(x, T - i\Delta t)) \qquad \hat{\sigma}_i^2 = \Delta t \sigma^2(i\Delta t). \tag{22}$$

These transition kernels constitute a hierarchical variational autoencoder of $L$ stochastic layers, whose marginal likelihood can be lower bounded by

$$\log p(x_L) \geq \mathbb{E}_q\left[\log p(x_0) + \sum_{i=0}^{L-1} \log \frac{p(x_{i+1}|x_i)}{q(x_i|x_{i+1})}\right] =: \mathcal{E}^L, \tag{23}$$

which we refer to as the discrete-time ELBO (DT-ELBO). The reconstruction error of the stochastic layer can be seen as some form of finite difference approximation to differentiation, which gives rise to $\nabla \cdot \mu$ in the CT-ELBO in the infinitesimal limit (as $\Delta t$ approaches 0). The regularization of $||a||^2$ pops up when we compare the difference between $\tilde{\mu}_i$ and $\hat{\mu}_i$ using the Gaussian reparameterization to compute the reconstruction error. We formalize this idea in the following theorem.

**Theorem 5** (**Consistency**). *Assume $\mu$, $\sigma$, $\sigma^{-2}$, $a$, $||a||^2$ and their derivatives up to the fourth order are all bounded and continuous, and that $\sigma$ is non-singular. Then $\mathcal{E}^L \to \mathcal{E}^\infty$ as $L \to \infty$.*

This theorem tells us that the CT-ELBO we derive for continuous-time diffusion models is not that different from the traditional ELBO, and that maximizing the CT-ELBO can be seen as training an infinitely deep hierarchical VAE. We present the proof in Appendix F, which formalizes the above intuition, using Taylor's theorem to control the polynomial approximation error, which will go to 0 as the step size $\Delta t$ vanishes when the number of layers $L$ increases to infinity.

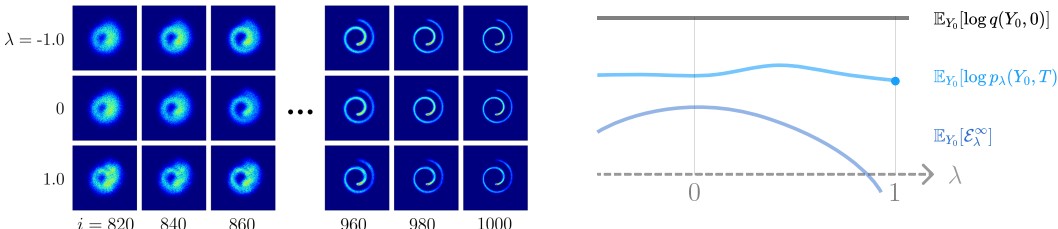

Figure 2: *Left*: Samples from plug-in reverse SDEs with different $\lambda$ values (rows). We use the same score function $\boldsymbol{s}_\theta$ trained on the Swiss roll dataset, and plug it into (27). For generation, we use the Euler Maruyama method with a step size of $\Delta t = 1/1000$. We visualize the samples for the $i$-th iterates (columns), which approximately represent the same marginal distribution when the score function is well trained. *Right*: Lower bound on the marginal likelihood of a continuum of plug-in reverse SDEs. The lower bound is optimized when the score matching loss is minimized, which will push up the entire dark blue curve.

## 6  Score-based generative modeling

Recall that our goal is to analyze the plug-in reverse SDE and draw connection to score matching. To this end, we reparameterize the generative (4) and inference (17) SDEs as

$$\mathrm{d}X = (gg^\top \boldsymbol{s}_\theta - f)\,\mathrm{d}t + g\,\mathrm{d}B_t \ \text{ and } \ \mathrm{d}Y = f\,\mathrm{d}s + g\,\mathrm{d}\hat{B}_s, \tag{24}$$

by letting $a = g^\top \boldsymbol{s}_\theta$, where the time variable is reversed ($T - t$) for the generative process, and forward in time ($s$) for inference. The ELBO (16) can be rewritten as

$$\mathcal{E}^\infty = \mathbb{E}_{Y_T}[\log p_0(Y_T)\,|\,Y_0 = x] - \int_0^T \mathbb{E}_{Y_s}\left[\frac{1}{2}||\boldsymbol{s}_\theta||^2_{gg^\top} + \nabla \cdot (gg^\top \boldsymbol{s}_\theta - f)\,\Big|\,Y_0 = x\right]\,\mathrm{d}s. \tag{25}$$

Comparing the integrand to the implicit score matching loss in Table 1, we immediately see that the network $\boldsymbol{s}_\theta$ approximates $\nabla \log q(y, s)$, the score function of the marginal density of $Y_s$. That is, *matching the score of $q(y, t)$ amounts to maximizing the lower bound on the marginal likelihood of the plug-in reverse SDE.*

Recently, Durkan & Song (2021)[6] also attempt to establish the equivalency between maximum likelihood and score matching, by showing the following relationship between the forward KL divergence and a weighted sum of score matching loss (aka the Fisher divergence):

$$D_{\mathrm{KL}}(q(y,0)||r(y,0)) = \frac{1}{2}\int_0^T \mathbb{E}_{q(\cdot, s)}\left[||\nabla \log r(Y_s, s) - \nabla \log q(Y_s, s)||^2_{gg^\top}\right]\,\mathrm{d}s, \tag{26}$$

where $r(y, s)$ is the density of $Y_s$ solving the same inference SDE with the initial condition $y_0 \sim r(\cdot, 0)$, assuming $q(y, T) = r(y, T)$. However, it is inaccurate to claim that score matching is equivalent to maximum likelihood. This is because if we simply let $r(y, 0) = p(y, T)$, *i.e.* the density of the generative SDE evaluated at $y$, $r(y, s)$ will not necessarily be the same as either $p(y, T - s)$ or $\boldsymbol{s}_\theta(y, s)$. This means the KL divergence is not equal to the integral of the weighted score matching loss $\mathbb{E}[\frac{1}{2}||\boldsymbol{s}_\theta - \nabla \log q||^2_{gg^\top}]$. In fact, the latter corresponds to a lower bound on the likelihood (the cross-entropy term of the KL) up to some constant, as equation (25) suggests.

More generally, we can apply our analysis to a family of plug-in reverse SDEs indexed by some parameter $\lambda \leq 1$:

$$\mathrm{d}X = \left(\left(1 - \tfrac{\lambda}{2}\right)g^2 \boldsymbol{s}_\theta - f\right)\mathrm{d}t + \sqrt{1 - \lambda}g\,\mathrm{d}B_t \ \text{ and } \ \mathrm{d}Y = \left(f - \tfrac{\lambda}{2}g^2\nabla \log q\right)\mathrm{d}s + \sqrt{1 - \lambda}g\,\mathrm{d}\hat{B}_s, \tag{27}$$

---

[5]We follow the Euler-Maruyama (EM) scheme. Other discretization scheme may also work; we leave that for future work.

[6]This refers to the v1 of the paper on arXiv. This version was later on replaced with a new version where they derived a similar bound as ours.

where we assume $g$ is diagonal for simplicity. We defer the formal discussion to Appendix G, but the essence is that this inference SDE induces the same marginal distribution as (1), and the generative SDE is its corresponding plug-in reverse. Equation (27) includes the original plug-in reverse SDE (24) and an equivalent ODE as special cases with $\lambda = 0$ and $\lambda = 1$. Denote its corresponding CT-ELBO by $\mathcal{E}_\lambda^\infty$. Specifically, (25) becomes $\mathcal{E}_0^\infty$. Then we have the following relationship.

**Theorem 6** (**Plug-in reverse SDE ELBO, abridged**). *For $\lambda < 1$,*

$$\mathbb{E}_{Y_0}[\mathcal{E}_\lambda^\infty] = \mathbb{E}_{Y_0}[\mathcal{E}_0^\infty] - \left(\frac{\lambda^2}{4(1-\lambda)}\right) \int_0^T \mathbb{E}_{Y_s}\left[\frac{1}{2}||\boldsymbol{s}_\theta(Y_s, s) - \nabla \log q(Y_s, s)||_{g^2}^2\right] \mathrm{d}s \qquad (28)$$

We state the full theorem in Appendix H, where we rearrange the terms to show that the average CT-ELBO of the $\lambda$-plug-in reverse SDE is also equivalent to the ISM loss, similarly to (25) but up to some multiplying and additive constants. The implication is that *while minimizing the score matching loss, we implicitly maximize the likelihood of a continuum of plug-in reverse SDEs which include the ODE as a limiting case ($\lambda \to 1$)*. See Figure 2 (right) for illustration. This suggests the likelihood of the equivalent ODE can be improved by minimizing the score matching loss, as the ODE's likelihood will be close to plug-in reverse SDEs with $\lambda \approx 1$, which explains the good likelihood of the equivalent ODE reported in Song et al. (2021). In practice, we can only estimate the ELBO of the case $\lambda = 0$ since otherwise there will be some constant we do not have access to, but their gradients can all be estimated via score matching.

## 6.1 Computational trade-off

Having a general framework for estimating the likelihood of diffusion processes allows us to compare a wide family of models, including continuous-time flows and plug-in reverse SDEs trained by score matching. We compare the two by measuring the negative ELBO throughout training to highlight their computation-estimation trade-off. We train the models on the Swiss roll toy data. For continuous-time flow, we set $\sigma = 0$, using the Hutchinson trace estimator following Grathwohl et al. (2018). The ELBO in this case is tight since $a$ will be penalized to be 0. We use the `torchdiffeq` library (Chen et al., 2018) for numerical integration for fairer comparison[7]. For plug-in reverse SDE, we train the drift network $a$ using SSM and DSM (for DSM the loss is weighted to reduce variance, which introduces some bias; see the next subsection). We use the variance-preserving inference SDE from Song et al. (2021), which allows us to sample $Y_s$ using a closed form formula, for $s$ sampled uniformly between $[0, T]$. The trained models are visualized in Figure 1, the learning curves presented in Figure 3.

From the learning curve figures, we see that neg-likelihood decreases rapidly for the continuous-time flow in terms of the number of parameter updates. But once the x-axis is normalized by runtime, the convergence speed becomes almost indistinguishable. This is because for continuous-time flows, numerical integration takes time, whereas for plug-in reverse SDEs, we train on a random time step $s$; that is, within a fixed amount of time the latter can make more parameter updates at the cost of noisier gradients. Note that both models have constant memory cost (wrt $T$ or $L$, the number of integration steps), so a large batch size can be used to reduce variance for training.

## 6.2 Bias and variance trade-off

The integral in equation (25) can be estimated by sampling $(Y_s, s)$, and using the Hutchinson trace estimator to estimate the divergence, which corresponds to implicit score matching. However, in practice the variance of this estimator is very high when the norm of the Jacobian $\nabla \boldsymbol{s}_\theta$ is large. Another popular approach is to use the denoising estimator (recall the identity from (3)),

$$\mathbb{E}_{Y_s}\left[\frac{1}{2}||\boldsymbol{s}_\theta(Y_s, s) - \nabla \log q(Y_s|Y_0)||_{gg^\top}^2 \,\middle|\, Y_0 = x\right]. \qquad (29)$$

The inference SDE is typically chosen so that $Y_s$ can be easily sampled, e.g. following $\mathcal{N}(\mu_s, \sigma_s^2)$, where $\mu_s$ and $\sigma_s$ are functions of $Y_0$ and $s$. In this case, if we reparameterize $Y_s = \mu_s + \sigma_s \epsilon$ where

---

[7]Black-box SDE solvers such as `torchsde` (Li et al., 2020) might not be optimized for the deterministic case, since their stochastic adjoint method scales $\mathcal{O}(L \log L)$ in time whereas deterministic numerical solvers are usually faster. This matters for our runtime comparison.

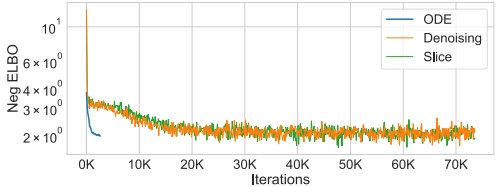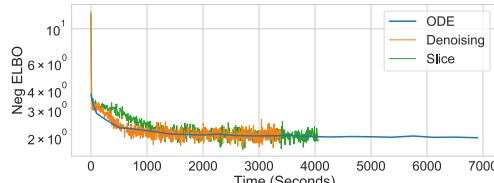

Figure 3: Neural ODE vs plug-in reverse SDE (denoising or slice score matching). The learning curves are presented as a function of iterations (left) and runtime (right) to emphasize the computational distinction between the two families of models.

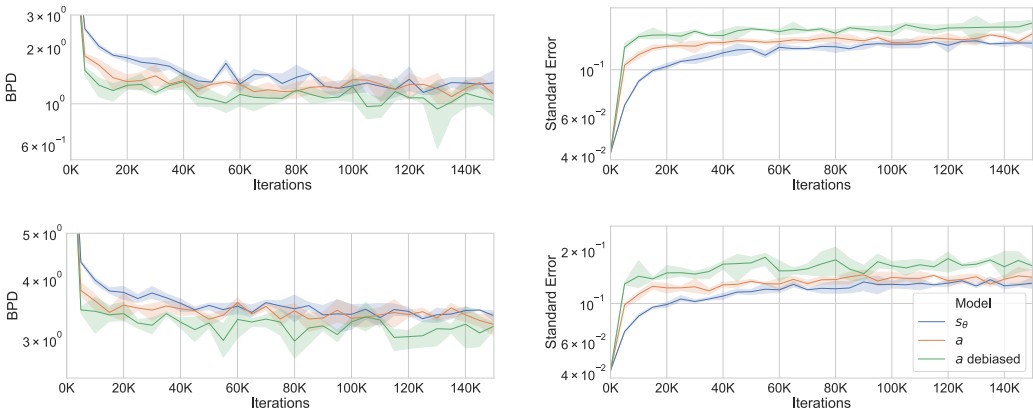

Figure 4: Likelihood estimation on MNIST (first row) and CIFAR10 (second row). $s_\theta$ and $a$ denote which model we parameterize. Y-axes are bits-per-dim and the standard error of BPD of the test set. The debiased curves improve upon the original biased gradient estimator (Song et al., 2021) since it maximizes a proper ELBO. Shaded area reflects the uncertainty estimated by 3 random seeds.

$\epsilon \sim \mathcal{N}(0, \mathbf{I})$, then the score becomes $\nabla \log q = -\frac{\epsilon}{\sigma_s}$. Since $\sigma_s \to 0$ as $s \to 0$, this estimator normally has unbounded variance. Song & Ermon (2019); Song et al. (2021) propose to remedy this by multiplying the DSM loss by $\sigma_s^2/g^2$ (assuming $g$ is a scalar for simplicity), so that the target has constant magnitude on average $\mathbb{E}[\frac{1}{2}||\sigma_s s_\theta + \epsilon||^2]$, which would result in a biased gradient estimate with much smaller variance. We can debias this estimator by sampling $s \sim q(s) \propto g^2/\sigma_s^2$. This ratio, however, is usually not normalizable in practice (as it integrates to $\infty$). As an alternative, we consider the following unnormalized density $\tilde{q}_\epsilon(s) = g^2(s_\epsilon)/\sigma_{s_\epsilon}^2$ for $s \in [0, s_\epsilon]$, and $\tilde{q}_\epsilon(s) = g^2(s)/\sigma_s^2$ for $s \in [s_\epsilon, T]$. We experiment with this debiased procedure by sampling $s \sim q_\epsilon \propto \tilde{q}_\epsilon$, for $f$ and $g$ chosen to be the variance-preserving SDE. $s_\epsilon$ is small so that the bias is negligible.

We train the model on MNIST (LeCun et al., 1998) and CIFAR10 (Krizhevsky et al., 2009). We present the learning curves and the standard error of the estimate of the ELBO in Figure 4. The lower bound is estimated using the Hutchinson trace estimator with $s$ sampled uniformly from $[0, T]$, with the same batch size, so the only thing that will affect the dispersion is the magnitude of $\nabla s_\theta$. Since smaller values of $s$ are more likely to be sampled under $q_\epsilon$, the debiased model will see samples with less perturbation more often. On the contrary, sampling $s$ uniformly will bias the model to learn from noisier data, causing the learned score to be smoother. We also experiment with parameterizing $s_\theta$ vs parameterizing $a$. We find the latter parameterization to be helpful since the relationship $s_\theta = g^{-1}a$ has the effect of negating the multiplier $\sigma_s$ in the reweighted loss, i.e. $\mathbb{E}[\frac{1}{2}||\frac{\sigma_s}{g}a + \epsilon||^2]$. This is similar to the noise conditioning technique introduced in Song & Ermon (2020).

## 7    Related work

**Diffusion-based generative models**    Our work lays a theoretical foundation for Song et al. (2021), which recognizes that conditional denoising score matching (Song & Ermon, 2019, 2020) and

discrete-time diffusion-based generative models (Sohl-Dickstein et al., 2015; Goyal et al., 2017; Ho et al., 2020) can be viewed as learning to revert an inference process (using the plug-in reverse SDE). Different from Ho et al. (2020), which shows the ELBO of discrete time diffusion process can be likened to DSM (Section 3.2 of the paper), we show that ISM loss naturally arises from the Fokker-Planck equation of the marginal density, via the Fenman-Kac representation and the Girsanov change of measure. This line of work has been successfully applied to modeling high dimensional natural images (Dhariwal & Nichol, 2021; Saharia et al., 2021), audio (Kong et al., 2020), 3D point cloud (Cai et al., 2020; Zhou et al., 2021), and discrete data (Hoogeboom et al., 2021).

**Time-reversal of diffusion processes**  Plenty of works have studied the reverse-time diffusion processes (2), including Anderson (1982); Föllmer (1985); Elliott & Anderson (1985); Haussmann & Pardoux (1986). These are different from our marginal-equivalent (reverse) processes (27) when $s_\theta = \nabla \log q$, since the latter is related by the marginals only.

**Score matching for energy-based models**  Besides the connection to diffusion models, score matching is also often used as a method for learning energy based models (EBM)— see Song & Kingma (2021) for a comprehensive review on useful techniques—. When used as an EBM, sampling from the conditional score model can be achieved by running the annealed Langevin diffusion (Neal, 2001), which is connected to free-energy estimaton in physics (Jarzynski, 1997), wherein the *path integral* is essentially a Feynman Kac representation.

**De Bruijn's identity**  To connect maximum likelihood and score matching, Durkan & Song (2021) shows that KL divergence can be represented as an integral of weighted Fisher divergence, generalizing the case of Lyu (2009) where the inference perturbation is a simple Brownian motion. This type of formulas fall into the category of de Bruijn's identity (Cover, 1999) for relative entropy. A similar differential form result can be found in Wibisono et al. (2017).

**Learning SDEs**  Tzen & Raginsky (2019); Li et al. (2020) also propose to learn a neural SDE by applying Girsanov's theorem. The key difference is that they treat the SDE entirely as a latent variable, with an additional emission probability, whereas we use the Feynman-Kac formula to directly express the marginal density as an expectation, side-stepping the need to smooth out the density using the emission probability (which will be a Dirac point mass in our case). In their case, the inference direction is the same as the generative direction, since they infer the latent SDE directly, whereas we apply Girsanov to the Feynman-Kac diffusion (opposite the generative direction). Xu et al. (2021) further apply neural SDE as an infinitely deep Bayesian neural network.

## 8 Conclusion and Discussion

In this work, we derive a general variational framework for estimating the marginal likelihood of continuous-time diffusion models. This framework allows us to study a wide spectrum of models, including continuous-time normalizing flows and score-based generative models. Using our framework, we show that performing score matching with a particular choice of mixture weighting is equivalent to maximizing a lower bound on the marginal likelihood of a family of plug-in reverse SDEs, of which the one used in Song et al. (2021) and the equivalent ODE are special cases. Empirically, we validate our theory by monitoring the ELBO while performing score matching, and discuss the implication of the choice of mixture weighting and the potential of debiasing via non-uniform sampling. We emphasize that our theory does not explain the impressive sample quality of this family of models, which is still an open research problem and we leave it for future work.

This work introduces a general framework to estimate the likelihood of diffusion-based models, which allows the parameters of both the generative and inference SDEs to be learned, using a numerical solver with constant memory cost (as per Remark 2). The training time can be reduced via the connection to score matching and the reverse-time parameterization (24) as long as $f$ and $g$ take a simple form (so that $Y_t$ can be sampled without numerical integration). For example, one can generalize the Ornstein-Uhlenbeck process to have non-linear (in time) $f$ and $g$, similar to the variance-presering SDE, by parameterizing the integral of $f$ using a monotone network (Sill, 1998; Kay & Ungar, 2000; Daniels & Velikova, 2010; Huang et al., 2018). This has been explored in a concurrent work by Kingma et al. (2021) in a different framework.

## Acknowledgements

We would like to thank David Kanaa, Ricky Chen, Simon Verret, Rémi Piché-Taillefer, Alexia Jolicoeur-Martineau, and Faruk Ahmed for giving their feedback on this manuscript. We would also like to thank the INNF+ 2021 reviewers and NeurIPS 2021 reviewers for their constructive suggestions, which help us improve the clarity of the paper. Chin-Wei is supported by the Google PhD fellowship.

We also acknowledge the Python community (Van Rossum & Drake Jr, 1995; Oliphant, 2007) for developing the tools that enabled this work, including numpy (Oliphant, 2006; Van Der Walt et al., 2011; Walt et al., 2011; Harris et al., 2020), PyTorch (Paszke et al., 2019), Matplotlib (Hunter, 2007), seaborn (Waskom et al., 2018), and SciPy (Jones et al., 2014).

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
