# A  Score matching losses

In this section, we prove the score matching loss identity for completeness. These proofs are adapted from Hyvärinen & Dayan (2005); Song et al. (2020); Vincent (2011) with slight modifications since we project the score onto the eigen-basis of $\Lambda(s)$. Recall the definition of the ESM loss

$$\mathcal{L}_{\text{ESM}} = \mathbb{E}_{Y_s}\left[\frac{1}{2}||\boldsymbol{s}_\theta(Y_s, s) - \nabla \log q(Y_s, s)||^2_{\Lambda(s)}\right] \tag{30}$$

where $Y_s \sim q(Y_s, s)$. Expanding the quadratic equation, we have

$$\mathcal{L}_{\text{ESM}} = \mathbb{E}_{Y_s}\left[\frac{1}{2}||\boldsymbol{s}_\theta(Y_s, s)||^2_{\Lambda(s)} - \boldsymbol{s}_\theta(Y_s, s)^\top \Lambda(s)\nabla \log q(Y_s, s) + \frac{1}{2}||\nabla \log q(Y_s, s)||^2_{\Lambda(s)}\right] \tag{31}$$

Moving $\mathcal{I}(q(Y_s, s))$ from the RHS to the LHS gives us

$$\mathcal{L}_{\text{ESM}} - \frac{1}{2}\mathcal{I}(q(Y)s, s)) = \mathbb{E}_{Y_s}\left[\frac{1}{2}||\boldsymbol{s}_\theta(Y_s, s)||^2_{\Lambda(s)} - \boldsymbol{s}_\theta(Y_s, s)^\top \Lambda(s)\nabla \log q(Y_s, s)\right] \tag{32}$$

## A.1  Implicit score matching

Now to draw connection to ISM, we apply integration by parts and the general Stokes' theorem (with mild regularity condition on $\boldsymbol{s}_\theta$) to the inner product term to obtain

$$\int q(y, s)\boldsymbol{s}_\theta(y, s)^\top \Lambda(s)\nabla \log q(y, s)\,\mathrm{d}y = \int \boldsymbol{s}_\theta(y, s)^\top \Lambda(s)\nabla q(y, s)\,\mathrm{d}y$$

$$= \int \nabla \cdot \left(q\Lambda^\top \boldsymbol{s}_\theta\right)\mathrm{d}y \nearrow^{0} - \int q\nabla \cdot \left(\Lambda^\top \boldsymbol{s}_\theta\right)\,\mathrm{d}y$$

$$= \mathbb{E}_{Y_s}\left[\nabla \cdot \left(\Lambda^\top \boldsymbol{s}_\theta\right)\right]$$

## A.2  Sliced score matching

For the SSM loss, use the Hutchinson trace estimator (Hutchinson, 1989) to replace the divergence operator, which is simply the trace of the Jacobian matrix.

## A.3  Denoising score matching

For DSM, similarly we first look at the inner product term

$$\int q(y, s)\boldsymbol{s}_\theta(y, s)^\top \Lambda(s)\nabla \log q(y, s)\,\mathrm{d}y = \int \boldsymbol{s}_\theta(y, s)^\top \Lambda(s)\nabla q(y, s)\,\mathrm{d}y$$

$$= \int \boldsymbol{s}_\theta(y_s, s)^\top \Lambda(y_s)\nabla \int q(y_s|y_0)q(y_0, 0)\,\mathrm{d}y_0\,\mathrm{d}y_s$$

$$= \int\int q(y_0, 0)\boldsymbol{s}_\theta^\top \Lambda\nabla q(y_s|y_0)\,\mathrm{d}y_s\,\mathrm{d}y_0$$

$$= \int\int q(y_0, 0)q(y_s|y_0)\boldsymbol{s}_\theta^\top \Lambda\nabla \log q(y_s|y_0)\,\mathrm{d}y_s\,\mathrm{d}y_0$$

$$= \mathbb{E}_{Y_0, Y_s}\left[\boldsymbol{s}_\theta^\top \Lambda\nabla \log q(Y_s|Y_0)\right]$$

where $q(y_s|y_0)$ denotes the conditional density of $Y_s$ given $Y_0$. Combining this with $\mathbb{E}_{Y_s}[||\boldsymbol{s}_\theta||^2_\Lambda]$, we have

$$\mathbb{E}_{Y_0, Y_s}\left[\frac{1}{2}||\boldsymbol{s}_\theta||^2_\Lambda - \boldsymbol{s}_\theta^\top \Lambda\nabla \log q(Y_s|Y_0)\right] = \mathbb{E}_{Y_0, Y_s}\left[\frac{1}{2}||\boldsymbol{s}_\theta - \nabla \log q(Y_s|Y_0)||^2_\Lambda\right] - \frac{1}{2}\mathbb{E}_{Y_0}[\mathcal{I}(q(y_s|Y_0))]$$

# B  Assumption of Feynman-Kac

**Assumption 1** (**Feynman-Kac**). *We assume the following: There exist some constants $B_h, B_v > 0$ and $p_h, p_v \geq 1$ such that $h \in C^0(\mathbb{R}^d)$ and $v \in C^{2,1}(\mathbb{R}^d \times [0, T])$ satisfy*

$$|h(y)| \leq B_h \left(1 + ||y||^{2p_h}\right) \ \text{ or } \ h(y) \geq 0 \tag{33}$$

$$\max_{0 \leq \varsigma \leq T} |v(y, \varsigma)| \leq B_v \left(1 + ||y||^{2p_v}\right) \tag{34}$$

# C  Mixture of continuous-time flows

Continuing the discussion in Remark 1, we analyze the limit of the determinant of the Jacobian of the finite approximation given the Brownian path: $x \leftarrow x + \mu(x, t)\Delta t + \sigma(t)\Delta B_i$. When the step size decreases to $0$, this should converge to the Itô integral. When $\Delta t$ is small enough, under the assumption that $\mu$ is uniformly Lipschitz, the finite approximation will be invertible for all steps. Then the determinant of the Jacobian of the overall transformation is just the product of determinant of each step:

$$
\begin{aligned}
\prod_i \det \left(\nabla \left(x + \mu(x,t)\Delta t + \sigma(t)\Delta B_i\right)\right) &= \prod_i \det \left(\mathbf{I} + \Delta t \nabla \mu\right) \\
&= \prod_i \left(1 + \Delta t \operatorname{Tr}(\nabla \mu) + \mathcal{O}(\Delta t^2)\right) \\
&= \exp\left(\sum_i \log\left(1 + \Delta t \nabla \cdot \mu + \mathcal{O}(\Delta t^2)\right)\right) \\
&= \exp\left(\sum_i \Delta t \nabla \cdot \mu + \mathcal{O}(\Delta t^2)\right) \\
&\rightarrow \exp\left(\int \nabla \cdot \mu\right) \ \text{ as } \Delta t \rightarrow 0
\end{aligned}
$$

This leads to the same derivation for the instantaneous change of variable formula for continuous time flow (Chen et al., 2018), but the argument of $\mu$ will be the solution of the Itô integral, instead of the solution of the deterministic dynamics only.

# D  Marginal density of diffusion models (general case)

In Section 3, we assume $\sigma$ is position-independent for simplicity. The general case of the Fokker-Planck equation can also be represented using the Feynmann-Kac formula. Following a similar conversion as in Table 2, we have

$$p(x, T) = \mathbb{E}\left[p_0(Y_T) \exp\left(\int_0^T -\nabla \cdot \mu(Y_s, T - s) + \sum_{i,j} \partial^2_{x_i, x_j} D_{ij}(Y_s, T - s)\, \mathrm{d}s\right) \ \middle| \ Y_0 = x\right]$$

where $Y_s$ solves

$$\mathrm{d}Y = -\tilde{\mu}(Y, T - s)\, \mathrm{d}s + \sigma(Y, T - s)\, \mathrm{d}B'_s$$

where $\tilde{\mu}(y, s)_i := \mu_i(y, s) - 2\sum_j \partial_{x_j} D_{ij}(y, s)$.

# E  One-dimensional explanation of Girsanov and variational inference

The Girsanov theorem (aka the Cameron–Martin-Girsanov theorem) describes how translation affects Wiener (Gaussian) measures. In Section 4, we deal with the infinite dimensional case, therefore demanding a formal measure-theoretic treatment. In this section, we use a one-dimensional case to illustrate how to interpret the Girsanov theorem and how we use it to derive the CT-ELBO, using the more familiar notion of probability density functions. Now imagine we do not have an

infinite-dimensional latent variable (*i.e.* the Brownian motion $B'$). Instead, imagine we have a one-dimensional latent variable $\epsilon'$ following a standard normal distribution. One can think about it as a VAE. This way, instead of having a classical Wiener measure (*i.e.* the distribution of Brownian motion) we would only need to deal with the standard Gaussian distribution, then $\mathbb{P}$ in (13) has density $p = \mathcal{N}(0, 1)$. Suppose $\mathbb{Q}$ also has density $q$ Then we can rewrite (13) using the more familiar density ratio

$$\mathbb{E}_q \left[ \log \frac{p}{q} + \cdots \,\middle|\, \cdots \right]$$

Recall $p(\epsilon') = \mathcal{N}(\epsilon'; 0, 1) = \frac{1}{\sqrt{2\pi}} e^{-\frac{1}{2}\epsilon'^2}$. If we translate this density by $a$ and let it be $q$, we have

$$q(\epsilon') = \frac{1}{2\pi} e^{-\frac{1}{2}(\epsilon'-a)^2}$$

This definition of $q$ gives us the density ratio

$$\frac{q(\epsilon')}{p(\epsilon')} = e^{a\epsilon' - \frac{1}{2}a^2}$$

Also, under the density $q$,

$$\hat{\epsilon} := \epsilon' - a$$

is again a standard normal random variable (which means $\epsilon'$ is a Gaussian random variable with mean $a$). Note the striking resemblance between the last two formulas and (14,15).

Now if we want to use this $q$ to perform inference as well as reparameterization, we simply just invert the standardization formula, by first sampling $\hat{\epsilon}$ from the standard normal distribution, and letting $\epsilon' = \hat{\epsilon} + a$. Under this reparameterization, the log-likelihood ratio $\log p/q$ in the ELBO becomes

$$-a\epsilon' + \frac{1}{2}a^2 = -a(\hat{\epsilon} + a) + \frac{1}{2}a^2 = -a\hat{\epsilon} - \frac{1}{2}a^2$$

Note that since $\hat{\epsilon}$ is the standard normal (under $q$), the first term is equal to $0$ in expectation. This derivation leads to the CT-ELBO in (16). See Section F for the formal proof.

# F   Proofs

**Theorem 3 (Continuous-time ELBO).** *Let $\mathbb{Q}$ be defined via the density (15). Then the RHS of (13) can be rewritten as*

$$\mathbb{E} \left[ -\frac{1}{2} \int_0^T ||a(\omega, s)||_2^2 \, ds + \log p_0(Y_T) - \int_0^T \nabla \cdot \mu \, ds \,\middle|\, Y_0 = x \right] =: \mathcal{E}^\infty, \qquad (16)$$

*where the expectation is taken wrt the Brownian motion $\hat{B}_s$, and $Y_s$ solves*[8]

$$dY = (-\mu + \sigma a) \, ds + \sigma d\hat{B}_s. \qquad (17)$$

*Proof.* By inverting the relationship (14), we have

$$dB'_s = d\hat{B}_s + a(\omega, s) \, ds$$

This allows us to reparameterize $Y_s$ as

$$dY = -\mu \, ds + \sigma \, dB'_s = -\mu \, ds + \sigma(d\hat{B}_s + a \, ds) = (-\mu + \sigma a) \, ds + \sigma + d\hat{B}_s$$

The log density can be written as

$$
\begin{aligned}
\log \frac{d\mathbb{P}}{d\mathbb{Q}} &= -\int_0^T a \cdot dB'_s + \frac{1}{2} \int_0^T ||a||^2 \, ds \\
&= -\int_0^T a \cdot (d\hat{B}_s + a \, ds) + \frac{1}{2} \int_0^T ||a||^2 \, ds \\
&= -\int_0^T a \cdot d\hat{B}_s - \frac{1}{2} \int_0^T ||a||^2 \, ds
\end{aligned}
$$

---

[8]Note that $\mu$ and $\sigma$ run backward in time from $T$, whereas $a$ runs forward.

Finally, since the first term is in expectation equal to zero (Øksendal, 2003, Theorem 3.2.1), we conclude the proof. ☐

**Theorem 4** (**Variational gap and optimal inference SDE**). *The variational gap can be written as*

$$\log p(x, T) - \mathcal{E}^\infty = \int_0^T \mathbb{E}\left[||a(\omega, s) - \sigma^\top \nabla \log p(Y_s, T - s)||^2\right] \, \mathrm{d}s. \tag{18}$$

*In particular, $\mathcal{E}^\infty = \log p(x, T)$ if and only if $a(\omega, s)$ can be written as $a(\omega, s) = a(Y_s(\omega), s)$ for almost every $s \in [0, T]$ and $\omega \in \Omega$, and $a(y, s) = \sigma^\top \nabla \log p(y, T - s)$ almost everywhere.*

*Proof.* To characterize the variational gap, we directly subtract the lower bound from the marginal likelihood:

$$\log p(x, T) - \mathcal{E}^\infty = \mathbb{E}\left[\log p(Y_0, T) - \log p(Y_T, 0) + \frac{1}{2}\int_0^T ||a(\omega, s)||_2^2 \, ds + \int_0^T \nabla \cdot \mu \, ds \,\middle|\, Y_0 = x\right]$$

The first two terms can be written as an integral

$$\log p(Y_0, T) - \log p(Y_T, 0) = -\int_0^T \mathrm{d} \log p(Y_s, T - s) \tag{35}$$

Using Itô's formula, we can rewrite the differential as

$$\mathrm{d} \log p(Y_s, T - s) = -\frac{\partial_s p(Y_s, T - s)}{p(Y_s, T - s)} \, \mathrm{d}s + \nabla \log p \cdot \mathrm{d}Y_s + \frac{1}{2} H_{\log p} : \mathrm{d}Y_s \mathrm{d}Y_s^\top$$

where $\mathrm{d}Y_s \mathrm{d}Y_s^\top = \sigma\sigma^\top \, \mathrm{d}s$.

After rearrangement, we have

$$\int_0^T \left[\frac{\partial_s p}{p} - \nabla \log p^\top(-\mu + \sigma a) - \frac{1}{2} H_{\log p} : \sigma\sigma^\top + \frac{1}{2}||a||^2 + \nabla \cdot \mu\right] \, \mathrm{d}s - \int_0^T \nabla \log p^\top \sigma \, \mathrm{d}\hat{B}_s \tag{36}$$

where the second term is equal to 0 in expectation.

Now using the Fokker Planck equation to expand $\partial_s p$, with further rearrangement and cancellation and by a final application of the conditional Fubini's theorem, we end us with the desired characterization of the gap.

☐

**Theorem 5** (**Consistency**). *Assume $\mu$, $\sigma$, $\sigma^{-2}$, $a$, $||a||^2$ and their derivatives up to the fourth order are all bounded and continuous, and that $\sigma$ is non-singular. Then $\mathcal{E}^L \to \mathcal{E}^\infty$ as $L \to \infty$.*

*Proof.* By definition of the log transitional distributions

$$\log p(x_{i+1}|x_i) = -\frac{d}{2}\log 2\pi - \log \det(\tilde{\sigma}_i) - \frac{1}{2}||x_{i+1} - \tilde{\mu}_i(x_i)||_{\tilde{\sigma}_i^{-2}}^2 \tag{37}$$

Using the definition of $\tilde{\mu}_i$, the quadratic term becomes

$$||x_{i+1} - x_i - \Delta t \mu(x_i, i\Delta t)||_{\tilde{\sigma}_i^{-2}}^2$$

Due the the Gaussian reparameterization (under $q$), we can write

$$\begin{aligned}
x_i &= \hat{\mu}_{i+1}(x_{i+1}) + \hat{\sigma}_{i+1}\epsilon \tag{38}\\
&= x_{i+1} + \Delta t\big(-\mu(x_{i+1}, (i+1)\Delta t)\\
&\qquad + \sigma((i+1)\Delta t)a(x_{i+1}, T - (i+1)\Delta t)\big) + \sqrt{\Delta t}\sigma((i+1)\Delta t)\epsilon
\end{aligned}$$

Plugging this into the quadratic term yields

$$
||\cdots||^2 = ||\Delta t(\mu(x_{i+1}, (i+1)\Delta t) - \mu(x_i, i\Delta t)) \tag{39}
$$
$$
- \Delta t\sigma((i+1)\Delta t)a(x_{i+1}, T-(i+1)\Delta t) - \sqrt{\Delta t}\sigma((i+1)\Delta t)\epsilon||^2
$$

We take care of the deviation in $\mu$ first, by taking the Taylor expansion around $(x_i, i\Delta t)$:

$$
\mu(x_{i+1}, (i+1)\Delta t) = \mu(x_i, i\Delta t) + \nabla\mu(x_i, i\Delta t)^\top(x_{i+1} - x_i) + \mathcal{O}(\Delta t) \tag{40}
$$

Note that the first order term wrt the time variable is also $\mathcal{O}(\Delta t)$, so it's absorbed into the remainder. Combining the last three identities, we have

$$
\frac{1}{2}||x_{i+1} - \tilde{\mu}_i(x_i)||^2_{\hat{\sigma}_i^{-2}} \tag{41}
$$
$$
= \frac{1}{2}\epsilon^\top\sigma^\top(\sigma\sigma^\top)^{-1}\sigma\epsilon + \Delta t\epsilon^\top\sigma^\top\nabla\mu^\top(\sigma\sigma^\top)^{-1}\sigma\epsilon + \frac{1}{2}\Delta t a^\top\sigma^\top(\sigma\sigma^\top)^{-1}\sigma a
$$
$$
+ o(\Delta t) + (\Delta t)^{1/2}\epsilon^\top\sigma^\top(\sigma\sigma^\top)^{-1}\sigma a
$$

Note that we've dropped the arguments of the functions for notational convenience. All the $\sigma$s in the denominator are $\sigma(i\Delta t)$. The $o(\Delta t)$ term can be neglected since it decays fast enough even though there are $L = 1/\Delta t$ of them. The last term is 0 in expectation since $\epsilon$ is Gaussian distributed. To take care of the first term (*), we turn to the log density of the inference model.

$$
\log q(x_i|x_{i+1}) = -\frac{d}{2}\log 2\pi - \log\det(\hat{\sigma}_{i+1}) - \frac{1}{2}||x_i - \hat{\mu}_{i+1}(x_{i+1})||^2_{\hat{\sigma}_{i+1}^{-2}} \tag{42}
$$
$$
= -\frac{d}{2}\log 2\pi - \log\det(\hat{\sigma}_{i+1}) - \frac{1}{2}||\hat{\sigma}_{i+1}\epsilon||^2_{\hat{\sigma}_{i+1}^{-2}} \tag{43}
$$

Comparing the third term with (*), we have

$$
\frac{1}{2}\epsilon^\top\sigma_{i+1}^\top\left((\sigma_{i+1}\sigma_{i+1}^\top)^{-1} - (\sigma_i\sigma_i^\top)^{-1}\right)\sigma_{i+1}\epsilon, \tag{44}
$$

where $\sigma_i := \sigma(i\Delta)$. Using the differential notation, in expectation, the above can be rewritten as

$$
\mathbb{E}\left[\frac{1}{2}\epsilon^\top\sigma^\top\left(\partial_t(\sigma\sigma^\top)^{-1}\right)\sigma\epsilon\right] dt = -\operatorname{tr}(\sigma^{-1}\partial_t\sigma)\, dt = -\partial_t\log\det(\sigma)\, dt, \tag{45}
$$

where we used Hutchinson's trace identity and Jacobi's formula. Therefore, the summation of the differences will converge to $\log\det(\sigma(0)) - \log\det(\sigma(T))$. This quantity will be negated by summing up the differences between the normalizing constants for all $L$ terms, which gives us $\log\det(\sigma(T)) - \log\det(\sigma(0))$, by the telescoping cancellation.

Now we only have two terms from the quadratic function, which will converge to

$$
\epsilon^\top\sigma^\top\nabla\mu^\top\sigma^{-\top}\epsilon\, dt + \frac{1}{2}||a||^2\, dt
$$

Using the trace identity again, and the fact that trace is similarity-invariant, we see that the above quantity is equal to

$$
\left(\nabla\cdot\mu + \frac{1}{2}||a||^2\right) dt
$$

in expectation. Now summing up all the layers, we can decompose the approximate error as

$$
\left|\mathbb{E}\left[\sum\log\frac{p}{q}\right] - \mathbb{E}\left[-\int\left(\nabla\cdot\mu + \frac{1}{2}||a||^2\right)\right]\right| \leq \left|\mathbb{E}\left[\sum\log\frac{p}{q} + \sum\left(\nabla\cdot\mu + \frac{1}{2}||a||^2\right)\Delta t\right]\right|
$$
$$
+ \mathbb{E}\left[\left|\sum\left(\nabla\cdot\mu + \frac{1}{2}||a||^2\right)\Delta t - \int\left(\nabla\cdot\mu + \frac{1}{2}||a||^2\right)\right|\right]
$$

As all the approximation errors are bounded and converge to 0 as $L \to \infty$, the first term goes to 0 by the *Dominated Convergence Theorem*. The assumption on the coefficients also guarantees the convergence in mean square error (Milshtein, 1975) of the Euler Maruyama scheme, which implies the second term goes to 0. The same applies to the last step for the prior term: $x_0 \to y(T)$ in $L^2$.

$\square$

# G   Equivalent SDEs

We use the following definition to formalize what we mean by equivalent SDEs

**Definition 1** (**Equivalent processes / SDEs**). *Let $Y_s$, $\tilde{Y}_s$ and $X_t$ be stochastic processes for $0 \leq s, t \leq T$. If $Y_s$ and $\tilde{Y}_s$ have the same distribution for all $s$, then they are said to be equivalent. If $X_t$ and $Y_{T-t}$ have the same distribution for all $t$, then we say $X_t$ is an equivalent reverse process. Two SDEs are equivalent if the processes they induce are equivalent. Two SDEs are equivalent reverse of each other if the processes they induce are equivalent reverse of one another.*

Note that when talking about the equivalency between SDEs, the dependency on an initial condition is implied.

In this section, we show how to construct a family of equivalent (reverse) SDEs. Let $Y_s$ be a diffusion process solving

$$\mathrm{d}Y = f \, \mathrm{d}s + g \, \mathrm{d}\hat{B}_s$$

We assume $g$ is position-independent and diagonal for simplicity. Let $\lambda \leq 1$, We can rearrange the Fokker-Planck equation to get

$$\partial_s q = -\nabla \cdot (fq) + \frac{1}{2}g^2 : H_q = -\nabla \cdot \left( \left( f - \frac{\lambda}{2}g^2 \nabla \log q \right) q \right) + \frac{1 - \lambda}{2}g^2 : H_q \quad (46)$$

Now let $f_\lambda := f - \frac{\lambda}{2}g^2 \nabla \log q$, and $g_\lambda := \sqrt{1 - \lambda}g$. Then the SDE $\mathrm{d}Y = f_\lambda \, \mathrm{d}s + g_\lambda \, \mathrm{d}\hat{B}_s$ has the same Fokker Planck equation as (46), which means the SDEs defined this way form a family of equivalent SDEs[9].

Now to construct an equivalent reverse SDE, we rearrange the Fokker Planck of this new SDE,

$$\partial_s q = -\nabla \cdot (f_\lambda q) + \frac{1}{2}g_\lambda^2 : H_q = -\nabla \cdot \left( (f_\lambda - g_\lambda^2 \nabla \log q) \, q \right) - \frac{1}{2}g_\lambda^2 : H_q \quad (47)$$

Now let $\mu_\lambda(x,t) := g_\lambda^2(x, T-t)\nabla \log q(x, T-t) - f_\lambda(x, T-t)$ and $\sigma_\lambda = g_\lambda(x, T-t)$. Then the SDE $\mathrm{d}X = \mu_\lambda \, \mathrm{d}t + \sigma_\lambda \, \mathrm{d}B_t$ with the initial condition $X_0 \sim q(\cdot, T)$ is an equivalent reverse SDE, since

$$\partial_t p = -\nabla \cdot (\mu_\lambda p) + \frac{1}{2}\sigma_\lambda^2 : H_p = \nabla \cdot \left( (f_\lambda - g_\lambda^2 \nabla \log q) \, p \right) + \frac{1}{2}g_\lambda^2 : H_p \quad (48)$$

is the time-reversal of (47). This also means there is a family of plug-in reverse SDEs parameterized by $\lambda$ and $\boldsymbol{s}_\theta$:

$$\mathrm{d}X = (g_\lambda^2 \boldsymbol{s}_\theta - f_\lambda) \, \mathrm{d}t + \sigma_\lambda \, \mathrm{d}B_t \quad (49)$$

$$= \left( \left( 1 - \frac{\lambda}{2} \right) g^2 \boldsymbol{s}_\theta - f \right) \, \mathrm{d}t + \sqrt{1 - \lambda}g \, \mathrm{d}B_t \quad (50)$$

The plug-in reverse SDE used by Song et al. (2021) corresponds to $\lambda = 0$, and the equivalent (plug-in) reverse ODE corresponds to $\lambda = 1$. See Figure 5 for the simulation.

# H   Score matching and plug-in reverse SDEs

In Section 6 we establish the connection between the score matching loss and the CT-ELBO of the plug-in reverse SDE for $\lambda = 0$. If we want to do the same for different values of $\lambda$, we need to make sure the generative and inference SDEs have the same diffusion coefficient (this is to make sure the Radon-Nikodym derivative is finite). In light of this, we define the following generative and inference pair

$$\mathrm{d}X = \left( \left( 1 - \frac{\lambda}{2} \right) g^2 \boldsymbol{s}_\theta - f \right) \, \mathrm{d}t + \sqrt{1 - \lambda}g \, \mathrm{d}B_t \quad \text{and} \quad \mathrm{d}Y = \left( f - \frac{\lambda}{2}g^2 \nabla \log q \right) \, \mathrm{d}s + \sqrt{1 - \lambda}g \, \mathrm{d}\hat{B}_s$$
$$(27)$$

Note that this is just the same equivalent SDE and equivalent (plug-in) reverse SDE from the Appendix G. We show that maximizing the ELBO of this family of plug-in reverse SDEs is also equivalent to performing score matching.

---

[9]Note that more generally the same would also hold if we let $\lambda$ be a time-dependent function.

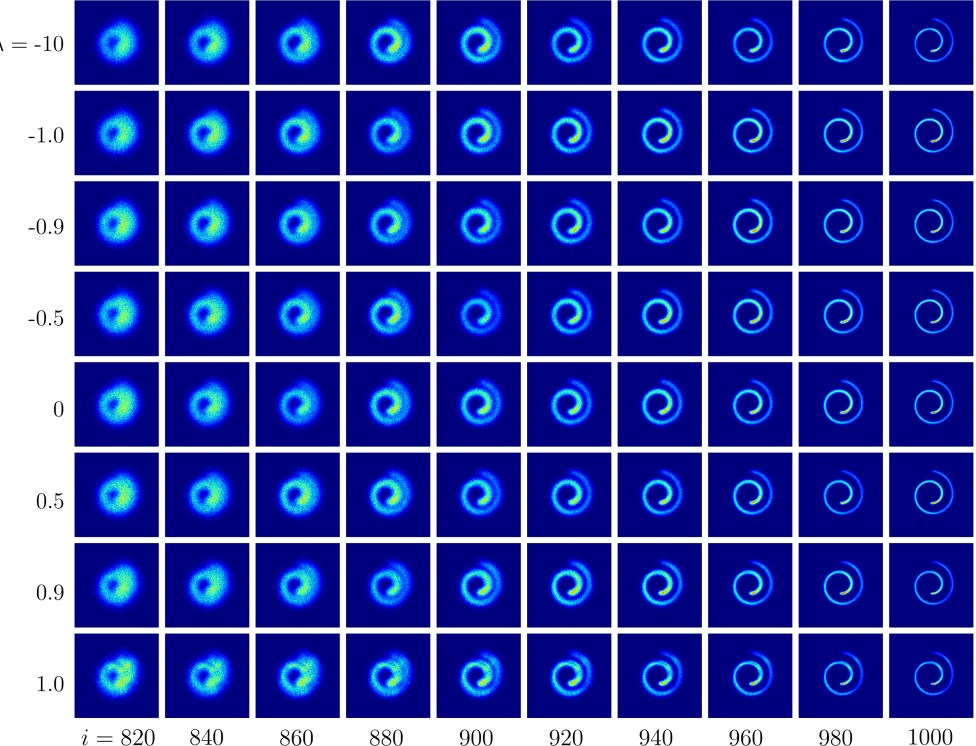

Figure 5: Samples from plug-in reverse SDEs with different $\lambda$ values (rows).

**Theorem 7 (Plug-in reverse SDE ELBO).** *Assume the generative and inference SDEs follow (27).
For $\lambda < 1$, then the CT-ELBO (denoted by $\mathcal{E}_\lambda^\infty$) can be written as*

$$\mathcal{E}_\lambda^\infty = \mathbb{E}_{Y_T}[\log p_0(Y_T) \,|\, Y_0 = x] - \int_0^T \left(1 - \frac{\lambda}{2}\right) \mathbb{E}_{Y_s}\left[\frac{1}{2}||\boldsymbol{s}_\theta||_{g^2}^2 + \nabla \cdot \left(g^2 \boldsymbol{s}_\theta - \left(\frac{2}{2 - \lambda}\right) f\right) \,\middle|\, Y_0 = x\right]$$

$$+ \frac{\lambda}{2} \mathbb{E}_{Y_s}\left[\frac{1}{2}||\boldsymbol{s}_\theta||_{g^2}^2 - g^2 \boldsymbol{s}_\theta^\top \nabla \log q(Y_s, s) \,\middle|\, Y_0 = x\right]$$

$$+ \frac{\lambda^2}{4(1 - \lambda)} \mathbb{E}_{Y_s}\left[\frac{1}{2}||\boldsymbol{s}_\theta - \nabla \log q(Y_s, s)||_{g^2}^2 \,\middle|\, Y_0 = x\right] \mathrm{d}s$$

*As a result, averaging the ELBO over the data distribution and applying the identity (3) yield*

$$\mathbb{E}_{Y_0}[\mathcal{E}_\lambda^\infty] = \mathbb{E}_{Y_T}[\log p_0(Y_T)] - \int_0^T \left(1 + \frac{\lambda^2}{4(1 - \lambda)}\right) \mathbb{E}_{Y_s}\left[\frac{1}{2}||\boldsymbol{s}_\theta||_{g^2}^2 + \nabla \cdot (g^2 \boldsymbol{s}_\theta)\right] \mathrm{d}s + \text{Const.}$$

$$(51)$$

$$= \mathbb{E}_{Y_0}[\mathcal{E}_0^\infty] - \left(\frac{\lambda^2}{4(1 - \lambda)}\right) \int_0^T \mathbb{E}_{Y_s}\left[\frac{1}{2}||\boldsymbol{s}_\theta(Y_s, s) - \nabla \log q(Y_s, s)||_{g^2}^2\right] \mathrm{d}s \qquad (52)$$

Before proving this theorem, we first make a few remarks. First, setting $\lambda = 0$, this ELBO will
reduce to (25). Second, (51) tells us that while matching the score, we implicitly maximize the
likelihood of the entire family of plug-in reverse SDEs. Third, (52) tells us that the average CT-
ELBO is maximized when $\lambda = 0$ (recall Figure 2). Lastly, the theorem excludes the case where
$\lambda = 1$, *i.e.* the equivalent ODE, since otherwise there will be a division-by-zero problem. But an
ODE can be seen as having $\lambda$ very close to 1, which will make the SDE effectively deterministic in
practice. This explains the low BPD of the equivalent plug-in ODE reported in Song et al. (2021).

*Proof.* Plugging (27) in (4) and (17), we get

$$\mu = \left(1 - \frac{\lambda}{2}\right) g^2 \boldsymbol{s}_\theta - f$$

$$\sigma = \sqrt{1-\lambda}\,g$$

$$a = \frac{1}{\sqrt{1-\lambda}}\left[(1-\lambda)\,g\boldsymbol{s}_\theta + \frac{\lambda}{2}g\,(\boldsymbol{s}_\theta - \nabla\log q)\right]$$

Then we have

$$\frac{1}{2}||a||_2^2 = \frac{1}{2(1-\lambda)}\left[(1-\lambda)^2\,||\boldsymbol{s}_\theta||_{g^2}^2 + (1-\lambda)\,\lambda g^2\boldsymbol{s}_\theta^\top\,(\boldsymbol{s}_\theta - \nabla\log q) + \frac{\lambda^2}{4}||\boldsymbol{s}_\theta - \nabla\log q||_{g^2}^2\right]$$

$$= \left(1 - \frac{\lambda}{2}\right)\frac{1}{2}||\boldsymbol{s}_\theta||_{g^2}^2 + \frac{\lambda}{2}\left(\frac{1}{2}||\boldsymbol{s}_\theta||_{g^2}^2 - g^2\boldsymbol{s}_\theta^\top\nabla\log q\right) + \frac{\lambda^2}{4(1-\lambda)}\frac{1}{2}||\boldsymbol{s}_\theta - \nabla\log q||_{g^2}^2$$

$$\nabla\cdot\mu = \left(1 - \frac{\lambda}{2}\right)\nabla\cdot\left(g^2\boldsymbol{s}_\theta - \left(\frac{2}{2-\lambda}\right)f\right)$$

Summing up these two parts gives us $\mathcal{E}_\lambda^\infty$. Under the expectation, we can rewrite $\mathbb{E}_{Y_s}[g^2\boldsymbol{s}_\theta^\top\nabla\log q] = -\mathbb{E}_{Y_s}[\nabla\cdot(g^2\boldsymbol{s}_\theta)]$ using the score matching loss identity (see Appendix A), to obtain the second part of the statement. $\square$

# I  Non-uniform sampling for debiasing

We perform non-uniform sampling to debias the denoising score matching loss weighted by $\sigma_s^2/g^2$, as discussed in subsection 6.2. We experiment with the variance-preserving SDE from Song et al. (2021) (originally from Ho et al. (2020)), whose drift and diffusion coefficients are

$$f(y,s) = -\frac{1}{2}\beta(s)y \tag{53}$$

$$g(y,s) = g(s) = \sqrt{\beta(s)} \tag{54}$$

where $\beta(s) = (\beta_{\max} - \beta_{\min})s + \beta_{\min}$, for some constants $\beta_{\max}$ and $\beta_{\min}$.

Solving the Fokker Planck of this SDE with a Dirac point mass as initial condition gives us a conditional Gaussian, whose variance is

$$\sigma_s^2 := \int_0^s g^2(s')ds' = \frac{1}{2}s^2(\beta_{\max} - \beta_{\min}) + s\beta_{\min} \tag{55}$$

Our goal is to sample from a density function proposal to $g^2/\sigma_s^2$ for most of the part. So for some small $s_\epsilon > 0$, we define the following unnormalized density

$$\tilde{q}_\epsilon(s) = \begin{cases} \frac{g^2(s_\epsilon)}{\sigma_{s_\epsilon}^2} & s \in [0, s_\epsilon) \\ \frac{g^2(s)}{\sigma_s^2} & s \in [s_\epsilon, T] \end{cases} \tag{56}$$

To simplify our notation, we let

$$\phi(s) := \log\left(\exp\left(\frac{1}{2}s^2(\beta_{\max} - \beta_{\min})s\beta_{\min}\right) - 1\right) \tag{57}$$

$$\varphi(u) := \log\left(1 + \exp\left(Zu + \phi(s_\epsilon) - \frac{g^2(s_\epsilon)}{\sigma_{s_\epsilon}^2}s_\epsilon\right)\right) \tag{58}$$

$$\tilde{\Phi}_\epsilon(s) := \begin{cases} \frac{g^2(s_\epsilon)}{\sigma_{s_\epsilon}^2}s & s \in [0, s_\epsilon) \\ \frac{g^2(s_\epsilon)}{\sigma_{s_\epsilon}^2}s_\epsilon + \phi(s) - \phi(s_\epsilon) & s \in [s_\epsilon, T] \end{cases} \tag{59}$$

where $\tilde{\Phi}_\epsilon$ is the cumulative function of the unnormalzied density. Evaluating it at $T$ gives us the normalizing constant $Z = \tilde{\Phi}_\epsilon(T)$, from which we obtain the CDF, $\Phi_\epsilon(s) = \frac{\tilde{\Phi}_\epsilon(s)}{Z}$, the pdf $q_\epsilon(s) = \frac{\tilde{q}_\epsilon(s)}{Z}$, and the inverse CDF that we need for sampling (using the inverse CDF transform):

$$
\Phi_\epsilon^{-1}(u) = \begin{cases} Z\frac{\sigma^2(s_\epsilon)}{g^2(s_\epsilon)}u & u \in \left[0, s_\epsilon \frac{g^2(s_\epsilon)}{Z\sigma^2_{s_\epsilon}}\right) \\ \frac{1}{\beta_{\max}-\beta_{\min}}\left(-\beta_{\min} + \sqrt{\beta^2_{\min} + 2\left(\beta_{\max} - \beta_{\min}\right)\varphi(u)}\right) & u \in \left[s_\epsilon \frac{g^2(s_\epsilon)}{Z\sigma^2_{s_\epsilon}}, 1\right] \end{cases} \tag{60}
$$

## J  Experiments

### J.1  MNIST and CIFAR 10

We use the variance preserving SDE described in Appendix I, with $\beta_{\min} = 0.1$, $\beta_{\max} = 20$, and $T = 1$. We use the same architecture following Ho et al. (2020) for the CIFAR10 experiment (which is a modified U-Net (Ronneberger et al., 2015)). For MNIST, we use 3 feature map resolutions (instead of 4) and reduce the number of channels from 128 to 32. Also we did not apply dropout.

For optimization, we use the Adam optimizer with a learning rate of 0.0001. We use minibatch size 128 for all experiments. We apply the standard uniform dequantization, and map the data to the real space using the logit transform (with a squeeze coefficient $\alpha = 0.05$ to avoid numerical instability). For CIFAR10, we additionally apply random horizontal flipping for regularization.

More details can be found in `https://github.com/CW-Huang/sdeflow-light`.