# OpenReview forum: "A Variational Perspective on Diffusion-Based Generative Models and Score Matching"
_NeurIPS.cc/2021/Conference — NeurIPS 2021 Spotlight_

### Official Review · Reviewer_99PV · 2021-07-15

**Rating:** 7
**Confidence:** 3

**Summary:**

The authors derive a general variational formulation for likelihood estimation in diffusion based generative modeling and draws an equivalence relation between maximising the associated continuous time (CT) ELBO and minimizing specific score-matching losses.

The authors also provide a more formal interpretation to “diffusion models can be viewed as a continuous limit of hierarchical VAEs” and interesting commentary on bias/ variance trade-off for weighted losses in score-matching.

**Limitations And Societal Impact:**

Limited societal impact, this paper is mainly theoretical.

**Main Review:**

Although previous attempts have been made to connect score-matching and likelihood objectives, this paper provides stronger results with more insight.
Formal interpretation to “diffusion models can be viewed as a continuous limit of hierarchical VAEs” is novel, though incremental by itself.
The results appear correct. Although not the focus of this work, experiments are limited however, see below.
Work is clear, though perhaps poorly organised. Perhaps re-stating existing theorems (Girsanov/ Feynman Kac) is unnecessary and the excess space may be used to provide more discussion on the paper’s contributions.
This work provides insights into score-matching losses used in practice and the link to likelihood, I believe this will be used and built upon by researchers.

Strengths:
Derive novel CT-ELBO and connect score matching to ELBO in continuous time. This is a general ELBO and allows comparison for various ODE and SDE reverse diffusion generative models.
Unlike previous work [2], the derived connection to specific score-matching losses is exact rather than an upper bound and provides insight into how various score-matching losses used in practice relate to likelihood and family of plug-in reverse SDEs. This formulation is elegant and provides better understanding.
Maximising ELBO in continuous time using ODE/ SDE integrators is computationally prohibitive, this paper derives an equivalency to a more scalable, albeit noisier approach using score-matching. I believe this will be of interest to practitioners as it gives a better understanding of score matching losses used in practice.
The authors formally confirm that limit of likelihood of discretized diffusion (i.e. infinite hierarchical VAE) matches continuous time likelihood
Interesting discussion on bias-variance trade-off via weighting scheme in score matching losses

Weaknesses:
The authors spend a significant proportion of the paper stating existing theorems (Girsanov, Keynman Kac representation), yet leave original Theorem 5 and discussion on plug-in reverse SDEs to the appendices.
Section 6 discussing the implications of main theoretical contribution is overly brief.
Although the bias/ variance trade-off from weighting is explored for images MNIST/ CelebA, the main contribution regarding likelihood based learning is only verified on low dimensional Swiss Roll data.
No experiments comparing numerical SDE integrators, just ODE.

Minor:
I do not think Durkan [2] claims score matching is equivalent to max likelihood as you state in line 195, just that score matching upper bounds likelihood and is a proxy.
Missing references:
Reverse SDE [1] (for instance)
Notation undefined:
Hessian H_p not defined in equation (7) line 88

[1] U. G. Haussmann. E. Pardoux. "Time Reversal of Diffusions." Ann. Probab. 14 (4) 1188 - 1205, October, 1986.
[2] Durkan, Conor, and Yang Song. "On maximum likelihood training of score-based generative models."


**Time Spent Reviewing:**

15

---

> ### Author Response · Authors · 2021-08-08
> **Official response to reviewer 99PV**
>
> We would like to thank you for your constructive feedback. We have largely improved the presentation of the paper (in terms of the main contribution) based on your comments.
>
> > re-stating existing theorems (Girsanov/ Feynman Kac) is unnecessary
>
> We respectfully disagree with you that stating Feynman-Kac and Girsanov's theorems is unnecessary. The reason why we stated them in the main text is that we chose to be more didactic in writing, as we do not believe most of our readers would be familiar with these results. We also believe these theorems can be useful tools to the probabilistic community that works on generative models in general.
>
> > the excess space may be used to provide more discussion on the paper’s contributions
>
> We agree the discussion in Section 6 could have been more thorough. We have taken your feedback into consideration and updated the draft to expand on the main contribution of the work. Specifically, we included an abridged version of Theorem 5 and discussed the lower bound on the likelihood of the entire family of marginal-equivalent plug-in reverse SDEs in more detail. Should the paper be accepted, we’d be granted an extra page to address reviewers’ suggestions and comments. Thank you for your feedback.
>
>
> > the main contribution regarding likelihood based learning is only verified on low dimensional Swiss Roll data
>
> The bias/variance paragraph also verifies likelihood based training: this is to verify using the proper likelihood weighting can further improve training (i.e. importance sampling), as opposed to the weighting originally proposed in Song et al., which is biased. The original weighting proposed by Song et al reduces variance but is biased. This experiment verifies debiasing this gradient estimator using non-uniform sampling of time can further improve the BPD.
>
> > No experiments comparing numerical SDE integrators, just ODE.
>
> As our goal is to make the point that for simple inference SDE, we can train the model via score matching (with randomly sampled time) to trade-off gradient noise for integration time, we only compare score-based training with neural ODE / FFJORD. We’ve added a new discussion paragraph in the conclusion section to mention that the more general setup is also feasible via numerical integration; we leave it for future work.
>
> > I do not think Durkan [2] claims score matching is equivalent to max likelihood
>
> Perhaps there has been some misunderstanding, as we quote below from their abstract
>
> “[...] we show that such an objective is equivalent to maximum likelihood for certain choices of mixture weighting.”
>
> The misunderstanding could be due to the fact that they have updated their arXiv with a new version with new results that is concurrent to ours. We intend to refer to the older version titled *”On” Maximum Likelihood Training of Score-Based Generative Models*. We’ve included an external link for reference: https://arxiv.org/pdf/2101.09258v1.pdf, and updated our draft to clarify this.
>
> > Missing references: Reverse SDE [1] (for instance)
>
> We’ve added the missing reference you suggested, thank you.
>
> > Notation undefined: Hessian H_p not defined in equation (7) line 88
>
> Hessian is defined in line 56.

---

> > ### Comment · Reviewer_99PV · 2021-08-25
> > **Thank you for your response**
> >
> > Thank you for your clarifications.
> > I had missed that your paper was referring to the previous version of "On Maximum Likelihood Training of Score-Based Generative Models" .
> > I am convinced by the changes proposed by the authors and will increase my score to 7.

---

### Official Review · Reviewer_xxp9 · 2021-07-16

**Rating:** 7
**Confidence:** 3

**Summary:**

The paper develops a variational framework for general diffusion processes. A continuous-time ELBO is developed that extends the ELBO of discrete-time diffusion processes. They use this to show that optimizing the score matching loss corresponds to maximizing a lower bound on the log marginal likelihood.

More specifically, the paper makes use of the Fokker-Planck/Kolmogorov forward equation along with the Feynman-Kac formula to express the marginal density of the generative diffusion process. Since the resulting expression is intractable, a change-of-measure is applied, introducing a variational approximation $\mathbb{Q}$ to $\mathbb{P}$. $\mathbb{Q}$ is chosen such that the Radon-Nikodym derivative $\frac{d\mathbb{Q}}{d\mathbb{P}}$ can be expressed using Girsanov theorem. From this, a continuous-time ELBO is obtained. Finally, through a specific choice of the drift coefficient $a$ in the inference SDE, a connection between the ELBO and score matching is established.

**Limitations And Societal Impact:**

In the checklist, the authors claim to describe limitations of their work, but with no reference to where. A dedicated paragraph to this could be beneficial. Potential negative societal impacts are not discussed.

**Main Review:**

This paper provides a variational framework for general continuous-time diffusion processes, which to the best of my knowledge is novel. The continuous-time ELBO is shown to consistently extend the usual ELBO for discrete-time diffusion processes, which is very elegant. Further for a specific choice of the inference SDE, the ELBO is shown to correspond (up to a constant) to a score-matching loss with a specific weighting ($\Lambda = gg^T$). This provides a connection between the training of continuous-time score-based diffusion models [1] and maximum likelihood training, concurrent with [2] which uses a different approach.

For the negatives: The experimental results are quite poorly presented. The plots in Fig. 3 and 4 and difficult to read. I would suggest 1) zooming in and 2) considering a log scaling also of the x-axis. Furthermore, a proper discussion of the results is missing (especially for MNIST/CIFAR-10), making it more difficult for the reader to get the take-aways from the experiments. Finally, details on the experiment setups appear to be missing.

This is a technically strong paper with somewhat heavy, but elegant theory. It adds to our theoretical understanding of score-based diffusion models and opens doors for further theoretical and experimental work. I would therefore like to see this paper accepted.

Minor questions/comments:
- A reference or appendix section on the relations in Eq. (3) would be useful.
- Line 71: You refer to $\mathcal{I}(q) = \mathbb{E}\left[ \| \nabla \log q \|_{\Lambda}^2 \right]$ as the Fisher information. Isn't the Fisher information commonly defined as the matrix $\mathcal{I}(q) = \mathbb{E}\left[ (\nabla \log q)(\nabla \log q)^T \right]$?

[1] Score-based generative modeling through stochastic differential equations (Song et al. 2020)
[2] Maximum Likelihood Training of Score-Based Diffusion Models (Song et al. 2021)

**Time Spent Reviewing:**

7

---

> ### Author Response · Authors · 2021-08-08
> **Official response to reviewer xxp9**
>
> We’d like to thank you for your positive feedback. We have updated our papers accordingly; see the detailed response below.
>
> * We have updated figure 3 to have larger linewidth and zoomed in on the Y-axis to improve figure 4. We also updated the description of the MNIST/CIFAR experiment to emphasize that the original weighting (used by Song et al 19’ and Song et al 21’) is biased and can be improved by sampling time non-uniformly to satisfy a proper ELBO. A description of the experimental setups is provided in Appendix H. We’ve also added a GitHub link in the updated draft for better reproducibility.
>
> * As per your suggestion, we have added a new appendix section for the proof of the score matching loss identity, i.e. Eq (3), to make the paper more self-contained.
>
> * Fisher information: $\mathcal{I}(q)$ is known as the total information in some literature. We agree this is potentially confusing. Thus we removed it and simply went by the definition to avoid confusion.

---

> > ### Comment · Reviewer_xxp9 · 2021-08-25
> > **Post-rebuttal**
> >
> > Thanks a lot for your reply. This is a strong paper that offers new theoretical insights on the promising class of score-based diffusion models. I am thus in favor of accepting the paper.

---

### Official Review · Reviewer_dz2y · 2021-07-18

**Rating:** 8
**Confidence:** 2

**Summary:**

This paper presents a framework for studying continuous diffusion models. Thanks to this it demonstrates the plug in generative model corresponds to a continuous version of the discrete diffusion models and is equivalent to minimising an ELBO on the likelihood of the model.
It does so by first transforming the change of density associated to the SDE into a second order linear differential equation and then expressing its solution as an expectation over another SDE. Solving this expectation is performed via variation inference on the Brownian paths and yields the continuous-time ELBO.
This CT-ELBO is then compared with its discrete time counter part. Finally it is compared to the score matching loss which demonstrates the soundness of such method for learning the generative model and continuous time diffusion models.


**Ethical Concerns:**

/

**Ethics Review Area:**

["I don’t know"]

**Limitations And Societal Impact:**

/

**Main Review:**

Originality:
To the best of my knowledge this paper present completely new results. The approach to derive the different expression is novel as well. Overall I think the proposed CT-ELBO could lead to many interesting future works for variation of the current continuous diffusion models.

Quality:
I have to admit that my background and the time I was able to spend on this review does not let me judge accurately the quality of the methods presented in this paper. However I think the methodology proposed by the authors to express the CT-ELBO is very nice and I don’t see fundamental flaws in those steps.
I think the authors could have expressed the role of lambda explicitly in the manuscript as it is discussed later in the experiments. I also believe authors should have discussed more thoroughly why the claims in Durkan and Song are not accurate.

Clarity:
Overall I think the paper is quite clear and make a really great at presenting complex materials into a comprehensive way.  However I also have to admit section 6 was a bit harder to follow on my side (except the first paragraph) and could maybe be improved.

Significance:
This work is very important as 1) It provides formal results for using plug-in score matching in diffusion generative models. 2) It provides the tools for creating new SDE based generative models.

Additional remarks:
• EQ 13: ds missing?
• I did not get how the first part of 15 disappear into 16.

**Time Spent Reviewing:**

5

---

> ### Author Response · Authors · 2021-08-08
> **Official response to reviewer dz2y**
>
> We’d like to thank you for the positive feedback. We have updated the manuscript to improve the clarity of the paper based on your comments.
>
> Specifically, we have expanded on the discussion in section 6 to talk about the family of lower bounds on plug-in reverse SDEs (i.e. Figure 5). The main takeaway of the paragraph on Durkan and Song is to not see $\log r$ as the same as the parametric score function $\mathrm{s}_\theta$. Here’s a counterexample in 1D: let $f$ and $g$ be the drift and diffusion of the standard Ornstein Uhlenbeck process (i.e. standard normal is its stationary distribution), and
>
> $$\mathrm{s}_\theta(x,t)=-100x$$
>
> Then $p(x, T)$ will be a centered Gaussian with a very small variance if $T$ is big (since the coefficient $-100$ magnifies the “zero-reverting” of the generative SDE). Set $q(y, 0)$ and $r(y, 0)$ to be exactly the same as the model $p(x,T)$. Then $D_{\text{KL}}(q(y,0) || r(y,0)) = 0 $ but $|| \mathrm{s}_\theta(y,s) - \nabla \log q(y, s) || $ won’t be zero (since $r(y,s)$ is the score of Gaussians with larger variance as $s$ increases, whereas $\mathrm{s}_\theta(x,t)$ does not change). This means score matching is not exactly the same as maximum likelihood.
>
> > role of lambda
>
> We have updated the draft to emphasize $\Lambda$ serves as a weighting function for the overall loss that will be used later on in Section 6.
>
> > EQ 13: ds missing?
>
> Yes. Thank you for catching it!
>
> > I did not get how the first part of 15 disappear into 16.
>
> If you mean why $\int a  \text{d} B’_s$ disappears, that’s because when we pull it down from the exponent, $\mathbb{E}[\int a  \text{d} \hat{B}_s]=0$ since the Itô integral is a Martingale starting from 0. This is discussed in Line 495-496 in the supplementary materials.

---

> ### Comment · Reviewer_dz2y · 2021-08-22
> **Post Rebuttal**
>
> I thank the authors for the clarification, they addressed all my small concerns. Moreover, in light of other reviews, I still believe this paper is very strong. I will thus increase my score to 8.

---

### Official Review · Reviewer_k6w1 · 2021-07-19

**Rating:** 7
**Confidence:** 4

**Summary:**

This paper rigorously shows how to calculate a variational lower bound on the marginal log density of a continuous time diffusion model. The bound contains implicit score matching, and it is calculated using Girsanov’s theorem from a Feynman-Kac representation of the marginal density obtained from the Fokker-Planck equation.

**Limitations And Societal Impact:**

Limitations are adequately addressed.

**Main Review:**

This is an excellent paper. I appreciate that the authors are rigorous for continuous time processes, and I like how the authors derive a marginal density without an emission probability.
My only concerns are regarding its framing in the context of prior work. My suggestion is to be more precise in describing the contributions as showing that *implicit* score matching arises from a variational lower bound on log likelihood, as opposed to existing and concurrent work showing that *denoising* score matching arises from a variational bound derived differently, starting with discrete time argument in DDPM (which, in section 7, should also be credited for showing that denoising score matching is learning to reverse an inference process).

If the goal of deriving implicit score matching is made clear throughout the paper, I think it will clarify to the reader the advantages of their method of deriving the variational lower bound as opposed to more traditional approaches, e.g. the approach from the original DPM paper, or modifying the approach from Tzen and Raginsky. And from this perspective, it is quite remarkable that the implicit score matching formula arises from Girsanov and Feynman-Kac, and I commend the authors on this finding.

It would be nice if the authors could discuss their ELBO in a more general setting, e.g. in which both forward and reverse processes have learnable drifts, or the setting of Tzen and Raginsky. How will training work in that case? Furthermore, can the authors discuss exactly when their method has advantages or disadvantages over other bounds, for example in lines 286-293? In the variational inference setting of Tzen and Raginsky, what does the authors’ proposed ELBO look like, and how does it compare to Tzen and Raginsky’s ELBO?

It appears that sections 3 and 4 use notation that refers to general SDEs (e.g. equation 4), but sections 2 and 6 use the same notation to refer to score-based generative modeling SDEs (e.g. equation 2). The change in notation is a bit confusing to the reader.


**Time Spent Reviewing:**

8

---

> ### Author Response · Authors · 2021-08-08
> **Official response to reviewer k6w1**
>
> Thank you for your appreciation and positive feedback. We’ve taken into account your comments to improve the presentation of the paper.
>
> Specifically, in the related work, we’ve further clarified how implicit score matching naturally arises from the probability flow of the diffusion model via the Girsanov change of measure, and contrasted it with the traditional derivation of DDPM which connects the ELBO of discrete time diffusion with denoising score matching.
>
> We also added an additional paragraph in the conclusion section to discuss the general setup where both inference and generative SDEs can be trained, via either 1) numerical integration, as Remark 2 suggests, or 2) simple but trainable parameterization of f and g. An example instantiation of 2) is via an extension of the Ornstein Uhlenbeck process where both the drift and diffusion coefficients are non-linear in time (generalizing the VP-SDE) and they can be parameterized as the derivative of a monotone network. This way, the forward / inference process will monotonically get closer to the standard Gaussian prior, and the sampling of $Y_s|Y_0$ still has an analytical form.
>
> The setting of Tzen and Raginsky requires an explicit emission probability, and the inference SDE runs the same direction as the generative SDE. In the case of “generative modeling” where one cares about the marginal density, the emission probability can be a Gaussian with a small variance. If the bound of Tzen and Raginsky is used, the “reconstruction error” can be large should the inference SDE deviate from the data point “x” too much due to the small variance. By contrast, our derivation directly deals with the evolution of the density of the generative SDE, which can be thought of as a stochastic version of the instantaneous change of variable (for neural ODE). This allows us to reliably track the change in density without the exploding signal due to the emission probability which might be detrimental to optimization because the initial condition for our inference SDE is “x”. Furthermore, the connection made with score matching (and the choice of linear f and g) allows us to sidestep numerical integration and the problem of numerical approximation error. More advanced variance reduction techniques can also be used to reduce the variance associated with the sampling of timestep during training.
>
> Choice of notation: Sections 3 & 4 are dedicated to the development of the general theory, whereas Sections 2 & 6 tie back to the plug-in reverse notation. We’ve added a footnote at the beginning of section 3 to explain the notation.

---

> > ### Comment · Reviewer_k6w1 · 2021-09-03
> > **Post rebuttal**
> >
> > Thanks to the authors for responding to my questions. The paper is enjoyable to read and I hope to see it accepted.

---

### Author Response · Authors · 2021-08-18
**Summary of changes**

We would like to thank all the reviewers for their unanimously positive ratings and their
constructive feedback. Below, we summarize a few major changes we’ve made to the manuscript. These changes are made to further improve clarity and emphasize the contributions made in this work.

* As per reviewer 99PV’s suggestion, we added an abridged version of theorem 5 to the discussion of the continuum of ELBOs in Section 6, and expanded on the discussion to make the contribution clearer.
* We derived and added a new theorem describing the variational gap in the general setup between Remarks 3 and 4, which justifies the use of a Markovian inference model $a(y, s)$. The theorem states that the gap is tight if and only if $a(y,s) = \sigma^\top\nabla \log p(y, T-s)$, which is the score of the generative SDE. And the gap is equal to
$$\log p(x, T) - \mathcal{E}^\infty =  \int_0^T \mathbb{E}[\lVert a(Y_s,s) - \sigma^\top\nabla \log p(Y_s, T-s) \rVert^2] \mathrm{d}s$$
* As per reviewer k6w1’s suggestion, we clarified how implicit score matching naturally arises from the probability flow of the diffusion model via the Girsanov change of measure, and contrasted it with the traditional derivation of DDPM which connects the ELBO of discrete time ELBO with denoising score matching. More specifically, the “cross-term”, i.e. the divergence of the parametric score, in ISM is linked to the divergence of the velocity field (the drift term) of the generative SDE, which tells us how much mass has entered / left an infinitesimal neighborhood around the particle evolving according to the generative SDE.
* We expanded the conclusion section to have a paragraph emphasizing the generality of our theory, and briefly discuss the possibility to train the inference SDE jointly with the generative SDE.

---

### Decision · Program_Chairs · 2021-09-27

**Decision:**

Accept (Spotlight)

**Comment:**

The main pros and cons that came up during reviews and discussions are:

pros:
* excellent paper with a elegant approach (k6w1, dz2y, xxp9, 99PV)
* new results opening up many interesting future works (dz2y, xxp9)


cons:
* framing in the context of prior work can be improved. (k6w1, dz2y)
* The advantages and disadvantages of proposed bound over other bounds could be clarified.  (k6w1)
* notation could be clarified. (k6w1)
* Section 6 can be improved in terms of clarity and can be expanded.(dz2y, 99PV)
* presentation of results can be improved. figures are hard to read and a discussion is missing. (xxp9)
* limited experimental results, although these are not the focus of the paper. (99PV)
* paper re-states too many existing theorems. Authors disagree, state it is important for a self-contained paper. (99PV)

The authors and reviewers engaged in discussions during the discussion period and most of the concerns raised by the reviewers were addressed. 3/5 reviewers raised their scores, leading to a unanimous recommendation of acceptance. The recommended decision by the meta-reviewer is to accept the paper.